# Learning from Label Proportions via Proportional Value Classification

**Tianhao Ma** [1]     **Wei Wang** [1,2]*     **Ximing Li** [3,2]     **Gang Niu** [2]     **Masashi Sugiyama** [2,1]

[1] The University of Tokyo, Japan

[2] RIKEN, Japan

[3] Jilin University, China

{matianhao2120,liximing86,gang.niu.ml}@gmail.com
wei.wang@riken.jp sugi@k.u-tokyo.ac.jp

## ABSTRACT

Learning from Label Proportions (LLP) aims to use bags of instances associated with the proportions of each label within the bag to learn an instance-level classifier. Proportion matching is a widely used strategy that aligns the average model outputs of all instances in a bag with the label proportions in order to induce the classifier. However, simply fitting the label proportions does not encourage discriminative instance-level predictions and may cause over-smoothing problems, resulting in poor classification performance. In this paper, we propose a novel LLP approach that can mitigate the over-smoothing problems with theoretical guarantees. Rather than fitting the label proportions directly, we treat them as targets for an auxiliary proportional value classification task to induce the target classifier. Our approach only requires the incorporation of an aggregation function after the classification layer. We also introduce an efficient computational approach with a divide-and-conquer strategy. Extensive experiments on various benchmark datasets and under different bag-generation strategies demonstrate that our approach achieves superior performance compared with state-of-the-art LLP methods. The code is publicly available at https://github.com/TianhaoMa5/ICLR2026_LLP-PVC.

## 1 INTRODUCTION

Deep learning has achieved remarkable success, but its effectiveness usually relies on large, accurately labeled datasets, which are impractical in many real-world scenarios. To address this problem, various weakly supervised learning paradigms have been explored in recent years, including semi-supervised learning (Zhu & Goldberg, 2009; Sohn et al., 2020), noisy label learning (Han et al., 2018; Wei et al., 2022), partial-label learning (Wang et al., 2022; 2025), and similarity-based classification (Hsu et al., 2019; Bao et al., 2022).

Learning from label proportions (LLP) is an emerging weakly supervised learning paradigm that has recently attracted much attention (Quadrianto et al., 2009; Busa-Fekete et al., 2023; Li et al., 2024b). In LLP, the training dataset consists of bags of instances annotated with the class proportions in each bag, while the instance-level labels are inaccessible to the learning algorithm. The goal of LLP is to learn an instance-level classifier that assigns correct labels to test instances, as in ordinary multi-class classification. Collecting bags of instances with label proportions is much easier and cheaper than collecting individual labels, which greatly reduces annotation costs. Furthermore, LLP provides an effective alternative when individual labels are inaccessible due to privacy concerns (Diemert et al., 2022). For example, in election polling, individual votes remain confidential, but aggregated results (such as regional support rates) are publicly available. By leveraging such bag-level data, LLP enables privacy-preserving analysis (e.g., predicting demographic voting trends) without accessing sensitive individual ballots. The need to learn from such *inexact supervision* arises widely in real-world applications, including high energy physics (Dery et al., 2017),

---

*Corresponding author.

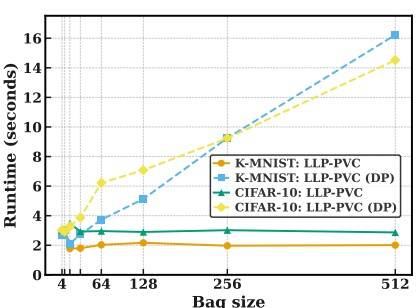

Figure 1: Curves of the average normalized entropy values of unlabeled instances in the training set and the test accuracy (Acc) for PM (Yu et al., 2014), and LLP-PVC (proposed) with a bag size of 128.

image classification (Li & Taylor, 2015; Ding et al., 2017), medical data analysis (Bortsova et al., 2018), and election prediction (Sun et al., 2017).

The most common strategy for LLP is *proportion matching* (Yu et al., 2014; Ardehaly & Culotta, 2017; Li et al., 2024b), which aligns the average value of the model predictions of all the instances in the bag with the label proportion. The proportion-matching strategy has been shown to be both concise and theoretically sound (Busa-Fekete et al., 2023; Li et al., 2024b) under the assumption that instances and labels are generated i.i.d. and then randomly grouped into bags. It has been widely adopted as the most common base loss in many LLP approaches, often combined with more sophisticated regularization techniques (Tsai & Lin, 2020; Ma et al., 2025). However, when applied to deep neural networks, proportion matching does not promote discriminative instance-level predictions and often suffers from the over-smoothing problem. A typical failure mode is that the model may collapse to a trivial solution where the model predictions of all instances approximate the same label proportion without distinction, which still fulfills proportion matching. Figure 1 shows the curves of the average entropy of training instances and the test accuracy for different LLP approaches. We can observe that the proportion matching approach similarly exhibits inferior model performance, with high entropy values throughout the training process. This suggests that the proportion matching strategy merely enforces that the average bag prediction matches the label proportions and does not encourage discriminative instance-level predictions. This will produce inaccurate instance-level predictions, resulting in poor classification performance.

To overcome this problem, we introduce a novel LLP approach with theoretical guarantees. Instead of using the label proportions as targets for direct fitting, we use them as targets for a novel classification problem. We observe that there exists a relationship between the posterior probabilities of the *discrete label proportion* and those of the instances in the bags. This motivates us to induce target classifiers by solving an auxiliary *proportional value classification (PVC)* task, where we only need to add an aggregation function after the classification layer of the original neural network to approximate the posterior probabilities of the proportional values. Figure 1 shows that the model predictions of our proposed approach have lower entropy values than proportion matching strategies. This shows that our approach mitigates over-smoothing problems and achieves better classification performance. Since the space of label sequences is very large, we use the one-versus-rest strategy to simplify the computational process. Furthermore, a divide-and-conquer method is adopted to reduce the computational complexity by using the fast Fourier transform. Compared with the dynamic programming (DP) method (Shukla et al., 2023) with a time complexity of $O(m^2)$, we achieve a time complexity of $O(m \log m)$.

Figure 2: Runtime comparison of LLP-PVC and LLP-PVC (DP). The values report the mean per-epoch runtime, computed by averaging over 30 epochs.

The superior efficiency of our proposed solution is clearly demonstrated in Figure 2. Besides, our approach enables end-to-end training with comparable memory and efficiency requirements, and is flexibly compatible with any neural network architectures and optimizers.

Our contributions can be summarized as follows:

- Methodologically, we propose a novel LLP approach that works by solving an auxiliary PVC problem to induce the classifier. We also introduce a divide-and-conquer method to improve computational efficiency.
- Theoretically, we prove that the model predictions induced by our approach mitigate over-smoothing problems under certain assumptions. We also prove the convergence rate of the proposed risk estimator by providing an estimation error bound.
- Empirically, our approach achieves superior classification performance over other state-of-the-art LLP methods across various datasets and different bag-generation strategies.

## 2 PRELIMINARY

**Ordinary multi-class classification.** Let $\mathcal{X} \subseteq \mathbb{R}^d$ denote a $d$-dimensional feature space, and $\mathcal{Y} = \{1, \ldots, q\}$ the label space with $q$ classes. Let $p(\boldsymbol{x}, y)$ be the joint density over $(\boldsymbol{x}, y) \in \mathcal{X} \times \mathcal{Y}$. The goal of multi-class classification is to induce a classifier $f : \mathcal{X} \mapsto \mathcal{Y}$, which minimizes the following *classification error*:

$$R_{0-1}(f) = \mathbb{E}_{p(\boldsymbol{x}, y)} \left[ \mathbb{I}(f(\boldsymbol{x}) \neq y) \right], \tag{1}$$

where $\mathbb{E}$ is the expectation, and $\mathbb{I}$ is an indicator function that returns 1 if the predicate holds; otherwise it returns 0. However, since the 0-1 loss is difficult to optimize, we usually replace the 0-1 loss with a *classification-calibrated* surrogate loss function $\mathcal{L}$ such as the cross-entropy loss. The *risk* can be then written as follows:

$$R(f) = \mathbb{E}_{p(\boldsymbol{x}, y)}[\mathcal{L}(f(\boldsymbol{x}), y)]. \tag{2}$$

**One-versus-rest (OVR) strategy.** The OVR strategy is a widely used multi-class classification method that decomposes the original multi-class problem into multiple binary classification subtasks (Rifkin & Klautau, 2004; Zhang, 2004). Specifically, the OVR risk is formulated as follows:

$$\bar{R}(f_1, f_2, \ldots, f_q) = \mathbb{E}_{p(\boldsymbol{x}, y)} \left[ \ell(f_y(\boldsymbol{x}), 1) + \sum_{k \in \mathcal{Y} \backslash \{y\}} \ell(f_k(\boldsymbol{x}), 0) \right], \tag{3}$$

where each $f_k : \mathcal{X} \mapsto [0, 1]$ is a binary classifier that outputs the predicted probabilities for the positive class, and distinguishes class $k$ from all the other classes and $\ell$ is a binary loss function. The OVR risk can also be expressed as the sum of the class-wise risks $\bar{R}_k(f_k)$:

$$\bar{R}(f_1, f_2, \ldots, f_q) = \sum_{k=1}^{q} \bar{R}_k(f_k),$$
$$\text{where } \bar{R}_k(f_k) = \mathbb{E}_{p(x, y)} \left[ \mathbb{I}\{y = k\} \ell(f_k(\boldsymbol{x}), 1) + \mathbb{I}\{y \neq k\} \ell(f_k(\boldsymbol{x}), 0) \right]. \tag{4}$$

**Learning from label proportions.** In LLP, we are given a training set consisting of bags of instances, where each bag is associated with a *label proportion* indicating the proportion of each label in the bag. Specifically, we are given a training set $\mathcal{D} = \{(\mathbf{B}^i, \boldsymbol{\alpha}^i)\}_{i=1}^{n}$ i.i.d. drawn from a joint density $p(\mathbf{B}, \boldsymbol{\alpha})$. Following Li et al. (2024b), $\mathbf{B} = [\boldsymbol{x}_1, \ldots, \boldsymbol{x}_m]$ denotes a bag consisting of $m$ unlabeled instances, where the *ground-truth label sequence* $\boldsymbol{y} = [y_1, \ldots, y_m]$ is inaccessible to the learning algorithm; and $\boldsymbol{\alpha} = [\alpha_1, \ldots, \alpha_q]$ denotes the label proportion associated with $\mathbf{B}$, where each element $\alpha_k$ is calculated as $\alpha_k = \sum_{j=1}^{m} \mathbb{I}(y_j = k)/m$ and we have $\sum_{k=1}^{q} \alpha_k = 1$. In this paper, we consider the traditional LLP setting where bags $\mathbf{B}$ are generated i.i.d., while allowing the instances within each bag to be non-i.i.d.. The goal of LLP is to induce an instance-level classifier $f : \mathcal{X} \mapsto \mathcal{Y}$ that can predict the labels of individual test instances.

## 3 METHODOLOGY

In this section, we first present our proposed approach via proportional value classification for learning from label proportions. Then, we present the theoretical analyses. Finally, we present an efficient solution of our proposed approach.

## 3.1 Proportional Value Classification

The goal of our proposed approach is to induce an instance-level classifier by training an auxiliary classifier to assign label proportion vectors to input bags. However, we find that the space of possible label proportion vectors and their corresponding ground-truth label sequences is too large to be computationally tractable. In particular, according to the *stars and bars theorem* in combinatorics (Brualdi, 2004), there are $\binom{m+q-1}{q-1}$ possible label proportion vectors for our problem. Moreover, for each possible label proportion vector $\boldsymbol{\alpha}$ there are a total of $(m!)/\prod_{k=1}^{q}(m\alpha_k)!$ possible label sequences. Therefore, we use the OVR strategy to simplify the computation process. Without loss of generality, we present how to induce the binary classifier for class $k \in \mathcal{Y}$ in Section 3. Specifically, examples from class $k$ are regarded as positive w.r.t. class $k$ and examples from all the other classes are regarded as negative. In this way, the unknown ground-truth label sequence $[y_1, \ldots, y_m]$ can be transformed as follows:

$$\tilde{\boldsymbol{y}}^k = \left[\tilde{y}_1^k, \ldots, \tilde{y}_m^k\right], \quad \text{where} \quad \tilde{y}_j^k = \begin{cases} 1, & y_j = k, \\ 0, & y_j \neq k. \end{cases} \tag{5}$$

Note that $\alpha_k$ in the original label proportion can also represent the proportion of positive examples after using the OVR strategy w.r.t. class $k$, i.e., $\alpha_k = \sum_{j=1}^{m} \tilde{y}_j^k/m$, and we have $\alpha_k \in \{0, 1/m, 2/m, \ldots, 1\}$. Because of the discrete nature of the proportional values, we introduce the PVC problem to train a classifier to classify the bag as one of the proportional values.

**Definition 1** (Proportional value classification (PVC))**.** Without loss of generality, we consider class $k$. Let $(\mathbf{B}, \tilde{\alpha}_k)$ denote a pair of random variables with a joint density $p(\mathbf{B}, \tilde{\alpha}_k)$, where $\mathbf{B}$ is the random variable of a bag defined in Section 2. Also,

$$\tilde{\alpha}_k = m\alpha_k + 1 \in \{1, 2, \ldots, m+1\} \tag{6}$$

is defined as the random variable of the corresponding *proportional value label*, which is determined by the label proportion $\boldsymbol{\alpha}$ of the bag $\mathbf{B}$. While proportional values naturally admit an ordinal formulation, in this work we adopt a simpler $(m+1)$-class multiclass approach: PVC trains a bag-level classifier $\mathbf{g}_k(\mathbf{B}) = [g_{k,1}(\mathbf{B}), \ldots, g_{k,m+1}(\mathbf{B})]$ that assigns each bag to one proportional value, with $g_{k,l}(\mathbf{B})$ denoting the probability of the $l$-th proportional value. The goal is achieved by minimizing the following PVC risk for class $k$:

$$\tilde{R}_k(\boldsymbol{g}_k) = \mathbb{E}_{p(\mathbf{B}, \tilde{\alpha}_k)} \left[ \mathcal{L}\left(\boldsymbol{g}_k(\mathbf{B}), \tilde{\alpha}_k\right)\right]. \tag{7}$$

In particular, PVC can be seen as a special case of multi-class classification, the difference being that each example for PVC is associated with a bag of instances instead of a single instance in the input space. Furthermore, by using the OVR strategy for each of the classes and summing up every loss value, the total risk is

$$\sum_{k=1}^{q} \tilde{R}_k(\boldsymbol{g}_k) = \sum_{k=1}^{q} \mathbb{E}_{p(\mathbf{B}, \tilde{\alpha}_k)} \left[ \mathcal{L}\left(\boldsymbol{g}_k(\mathbf{B}), \tilde{\alpha}_k\right)\right]. \tag{8}$$

By transforming the original label proportions to proportional values defined in Definition 1 for the original training set, we can obtain a new training set $\tilde{\mathcal{D}}_k = \{(\mathbf{B}^i, \tilde{\alpha}_k^i)\}_{i=1}^n$ for class $k$. Since the data generation process of $\tilde{\mathcal{D}}_k$ is in accordance with that of the random variables $(\mathbf{B}, \tilde{\alpha}_k)$, we can consider $\tilde{\mathcal{D}}_k$ as sampled i.i.d. from $p(\mathbf{B}, \tilde{\alpha}_k)$. We generate the data for each class and conduct *empirical risk minimization* to approximate Eq. (8) as

$$\frac{1}{n} \sum_{k=1}^{q} \sum_{i=1}^{n} \mathcal{L}\left(\boldsymbol{g}_k\left(\mathbf{B}^i\right), \tilde{\alpha}_k^i\right). \tag{9}$$

Before introducing how to compute $\boldsymbol{g}_k$, we first clarify an implicit relationship between the posterior probabilities of the proportional values and those of the instances in the bags.

## 3.2 Bridging the Posterior Probabilities of Proportional Values and Instance-Level Labels

Obviously, there are $2^m$ possible binary label sets corresponding to the labels of the instances in a bag. We introduce a posterior probability vector $\boldsymbol{p}\left(\tilde{\boldsymbol{y}}^k|\mathbf{B}\right) \in [0,1]^{2^m}$, where the $r$-th dimension represents the posterior probability of the $r$-th possible label sequence. Let $\mathrm{Bin}(r, m) =$

$\left[b_1^{(r)}, \ldots, b_m^{(r)}\right]$ denote the *m-bit binary representation* of a decimal number $r$. Here, $b_m^{(r)}$ is the least significant bit while $b_1^{(r)}$ is the most significant bit. For convenience, we consider the $r$-th possible label sequence to be the binary representation of $(r-1)$. For example, we have $\mathrm{Bin}(3,3) = [0,1,1]$, which is the 4-th possible label sequence within the 8 possible label sequences in total. Specifically, for the $r$-th label sequence, we assign

$$\tilde{\boldsymbol{y}}^k = \left[\tilde{y}_1^k, \ldots, \tilde{y}_m^k\right] = \mathrm{Bin}(r-1, m) = \left[b_1^{(r-1)}, \ldots, b_m^{(r-1)}\right], \tag{10}$$

which indicates that the label of each instance equals the corresponding bit of the binary representation of $(r-1)$. Then, based on the independence of the instances in the bag, the $r$-th dimension of $\boldsymbol{p}\left(\tilde{\boldsymbol{y}}^k | \mathbf{B}\right)$ can be computed as

$$p\left(\tilde{\boldsymbol{y}}^k = \mathrm{Bin}(r-1, m)|\mathbf{B}\right) = \prod_{j=1}^m p(\tilde{y}_j^k = b_j^{(r-1)}|\boldsymbol{x}_j), r \in \{1, 2, \ldots, 2^m\}. \tag{11}$$

Furthermore, we introduce a matrix $\mathbf{Q} = [q_{r,l}]_{2^m \times (m+1)} \in [0,1]^{2^m \times (m+1)}$. Here, each element $q_{r,l} = p(\tilde{\alpha}_k = l|\tilde{\boldsymbol{y}}^k = \mathrm{Bin}(r-1, m))$ indicates the conditional probabilities of proportional values given a binary label sequence. For ease of presentation, we introduce the Hamming weight function $\mathrm{HW}(N)$ which returns the number of the bits of ones in a binary number $N$ (Wei, 2002). Then, $q_{r,l}$ can be determined as

$$\forall r \in \{1, 2, \ldots, 2^m\}, l \in \{1, 2, \ldots, m+1\}, q_{r,l} = \begin{cases} 1, & l = \mathrm{HW}\left(\mathrm{Bin}(r-1, m)\right) + 1; \\ 0, & \text{otherwise.} \end{cases} \tag{12}$$

Based on the above definitions, we introduce the following lemma.

**Lemma 1.** *The posterior probability of the proportional value $p(\tilde{\alpha}_k | \mathbf{B})$ and the posterior probabilities of the ground-truth labels of the instances $p(\tilde{y}_j^k | \boldsymbol{x}_j)$ in the bag satisfy*

$$p(\tilde{\alpha}_k = l | \mathbf{B}) = \sum_{r=1}^{2^m} q_{r,l} \prod_{j=1}^m \left( p\left(\tilde{y}_j^k = 1 | \boldsymbol{x}_j\right)^{b_j^{(r-1)}} p\left(\tilde{y}_j^k = 0 | \boldsymbol{x}_j\right)^{\left(1 - b_j^{(r-1)}\right)} \right). \tag{13}$$

The proof can be found in Appendix B.1. Lemma 1 bridges the posterior probabilities of the proportional values and those of the ground-truth labels of instances. We provide an illustrative example to facilitate the understanding of the relationship of posterior probabilities.

**Example 1.** Consider a bag $\mathbf{B} = [\boldsymbol{x}_1, \boldsymbol{x}_2]$ for $m = 2$. Then, there are 4 possible label sequences $\{[0,0], [0,1], [1,0], [1,1]\}$ and 3 possible proportional values $\{1, 2, 3\}$. Based on Eqs. (11 and 12), we have $\mathbf{Q}$ and the following equations:

$$\mathbf{Q} = \begin{bmatrix} 1 & 0 & 0 \\ 0 & 1 & 0 \\ 0 & 1 & 0 \\ 0 & 0 & 1 \end{bmatrix}, \quad \begin{cases} p(\tilde{\alpha}_k = 1|\mathbf{B}) = p\left(\tilde{\boldsymbol{y}}^k = [0,0]|\mathbf{B}\right) = p\left(\tilde{y}_1^k = 0|\boldsymbol{x}_1\right) p\left(\tilde{y}_2^k = 0|\boldsymbol{x}_2\right); \\ p(\tilde{\alpha}_k = 2|\mathbf{B}) = p\left(\tilde{\boldsymbol{y}}^k = [0,1]|\mathbf{B}\right) + p\left(\tilde{\boldsymbol{y}}^k = [1,0]|\mathbf{B}\right) \\ \qquad = p\left(\tilde{y}_1^k = 0|\boldsymbol{x}_1\right) p\left(\tilde{y}_2^k = 1|\boldsymbol{x}_2\right) + p\left(\tilde{y}_1^k = 1|\boldsymbol{x}_1\right) p\left(\tilde{y}_2^k = 0|\boldsymbol{x}_2\right); \\ p(\tilde{\alpha}_k = 3|\mathbf{B}) = p\left(\tilde{\boldsymbol{y}}^k = [1,1]|\mathbf{B}\right) = p\left(\tilde{y}_1^k = 1|\boldsymbol{x}_1\right) p\left(\tilde{y}_2^k = 1|\boldsymbol{x}_2\right). \end{cases}$$

Example 1 not only provides a more intuitive understanding of the relationship of the posteriors of the proportional values and instance-level labels, but also further verifies Lemma 1. Based on the above observations, we use a binary classifier $f_k$ to approximate the instance-level posterior probabilities $p\left(\tilde{y}_j^k = 1 | \boldsymbol{x}_j\right)$ by using the sigmoid function. Then, we attempt to approximate the posterior probabilities of proportional values $p(\tilde{\alpha}_k | \mathbf{B})$ by using the bag-level classifier $\boldsymbol{g}_k(\mathbf{B}) = \left[g_{k,1}(\mathbf{B}), g_{k,2}(\mathbf{B}), \ldots, g_{k,m+1}(\mathbf{B})\right]$, which is calculated as

$$g_{k,l}(\mathbf{B}) = \sum_{r=1}^{2^m} q_{r,l} \prod_{j=1}^m \left( f_k\left(\boldsymbol{x}_j\right)^{b_j^{(r-1)}} \left(1 - f_k\left(\boldsymbol{x}_j\right)\right)^{\left(1 - b_j^{(r-1)}\right)} \right), l \in \{1, 2, \ldots, m+1\}. \tag{14}$$

Here, $q_{r,l}$ and $b_j^{(r-1)}$ are constants that can be calculated in advance before the whole training process. Therefore, our approach does not introduce any hyperparameters. By taking the instantiation of $\boldsymbol{g}_k$ into Eq. (8), we can relate the proposed classification risk with the introduced binary classifiers. Let $\tilde{R}(f_1, f_2, \ldots, f_q)$ and $\hat{\tilde{R}}(f_1, f_2, \ldots, f_q)$ denote the classification risk in Eq. (8) and empirical risk in Eq. (9) associated with binary classifiers, respectively. Then, we can train binary classifiers $f_k, k \in \{1, 2, \ldots, q\}$, by minimizing $\hat{\tilde{R}}(f_1, f_2, \ldots, f_q)$ in Eq. (9) with datasets $\tilde{\mathcal{D}}_k$. The algorithmic details are summarized in Algorithm 1. We share the representation learning layers and use specific classification layers for different labels, which allows simultaneous training of all binary classifiers effectively and efficiently.

---

**Algorithm 1** LLP-PVC

---

**Require:** Binary classifiers $f_1, \ldots, f_q$ with shared representation layers; LLP dataset $\mathcal{D}$; maximum epochs $T_{\max}$; maximum iterations $I_{\max}$.
**Ensure:** Trained classifiers $f_1, \ldots, f_q$.
1: **for** $t = 1, 2, \ldots, T_{\max}$ **do**
2:     **Shuffle** $\mathcal{D}$;
3:     **for** $j = 1, \ldots, I_{\max}$ **do**
4:         **Fetch** mini-batch $\mathcal{D}_j$ from $\mathcal{D}$;
5:         **Forward** $\mathcal{D}_j$ and get the outputs of $f_1, \ldots, f_q$;
6:         **Compute** the output of $g$ based on Algorithm 2;
7:         **Update** $g$, which induces an update on $f_1, \ldots, f_q$;
8:     **end for**
9: **end for**

---

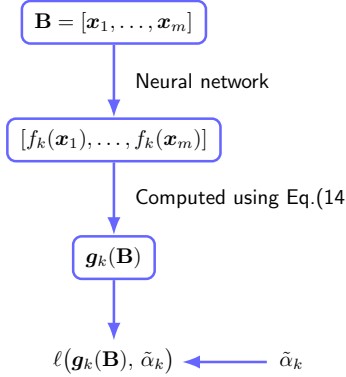

Figure 3: Pipeline of LLP-PVC.

### 3.3 THEORETICAL ANALYSIS

**Mitigation of over-smoothing problems.** We show that the induced classifier can mitigate over-smoothing issues under certain assumptions. First, we show that the output of the bag-level multi-class classifier $\boldsymbol{g}_k$ and the set of outputs of the instance-level binary classifier $f_k$ have a *one-to-one correspondence*.

**Theorem 1.** *The bag-level classifier $\boldsymbol{g}_k$ is an* invertible aggregation *of the instance-level binary classifier $f_k$ in the following sense: The model output of the bag-level classifier $\boldsymbol{g}_k(\mathbf{B})$ calculated by Eq. (14) uniquely determines the set of the instance-level model outputs $\{f_k(\boldsymbol{x}_j)\}_{j=1}^m$ for $\mathbf{B}$.*

*Proof Sketch (the full proof is given in Appendix B.3).* Firstly, we introduce a binomial for $f_k(\boldsymbol{x}_j), j \in \{1, \ldots, m\}$:

$$F_{k,j}(z) = f_k(\boldsymbol{x}_j)z + \big(1 - f_k(\boldsymbol{x}_j)\big). \tag{15}$$

Then, we define an $m$-order polynomial as the product of all $m$ binomials:

$$G_k(z) = \prod_{j=1}^m F_{k,j}(z) = \prod_{j=1}^m \Big(f_k(\boldsymbol{x}_j)z + (1 - f_k(\boldsymbol{x}_j))\Big) = \sum_{t=0}^m c_{k,t}\, z^t, \tag{16}$$

where $c_{k,t}$ is the coefficient of $z^t$. We have the following lemma.

**Lemma 2.** *For $t \in \{0, 1, \ldots, m\}$, the coefficient $c_{k,t}$ in Eq. (16) coincides with the $(t+1)$-th dimension of the model output of the bag-level classifier $\boldsymbol{g}$, i.e., $c_{k,t} = g_{k,t+1}(\mathbf{B})$.*

The proof is given in Appendix B.2. Lemma 2 indicates that when the bag-level model output $\boldsymbol{g}_k(\mathbf{B})$ is given, the $m$-order polynomial $G_k(z)$ can be determined. Notably, $G_k(z)$ has $m$ roots $\{1 - 1/f_k(\boldsymbol{x}_j)\}_{j=1}^m$, and each root has a one-to-one correspondence to the model output of the instance-level classifier for an instance. When $G_k(z)$ is uniquely determined, the roots are uniquely determined correspondingly. Then, we can prove by contradiction that the set of model outputs of the instance-level classifier is also determined uniquely since an $m$-order polynomial at most has $m$ different roots. □

Theorem 1 illustrates the one-to-one relationship between instance-level model outputs and bag-level model outputs based on Eq. (14). Once the model output of the bag-level classifier is given, then the set of the model outputs of the instance-level classifier can be determined. Let $\boldsymbol{g}_k^* = \arg\min_{\boldsymbol{g}_k} \tilde{R}_k(\boldsymbol{g}_k)$ denote the optimal classifier that minimizes Eq. (7). Then, the following theorem holds.

**Theorem 2.** *Assume that the cross-entropy loss or the mean squared error (MSE) loss is used to instantiate $\mathcal{L}$, and that the hypothesis class $\mathcal{G}$ for learning $\boldsymbol{g}_k$ is sufficiently flexible so that $\boldsymbol{g}_k^* \in \mathcal{G}$. Let $f_k^*$ denote the optimal binary classifier induced by $\boldsymbol{g}_k^*$, then we have $\{f_k^*(\boldsymbol{x}_j)\}_{j=1}^m = \{p(\tilde{y}_j^k = 1 \mid \boldsymbol{x}_j)\}_{j=1}^m$. If we further assume that the instance labels are deterministic, then it follows that $f_k^*(\boldsymbol{x}_j) \in \{0, 1\}$.*

The proof can be found in Appendix B.4. According to Theorem 2, when the labels are deterministic, the outputs of the minimizers will be 0 or 1 if the proposed risk in Eq. (7) is minimized. This suggests that our approach mitigates the over-smoothing problems of proportion matching strategies, where the model outputs of different instances in the bag may indiscriminately fit the given label proportion values. Furthermore, we observe that the set of outputs of $f_k^*$ coincides with the set of posterior probabilities of class $k$ for the instances. We leave for future work the investigation of the role of this property in training instance-level classifiers.

**Estimation error bound.** Let $(\hat{f}_1, \hat{f}_2, \ldots, \hat{f}_q) = \arg\min_{f_1, f_2, \ldots, f_q} \hat{\tilde{R}}(f_1, f_2, \ldots, f_q)$ denote the optimal classifiers by minimizing Eq. (9). Let $d$ denote the pseudo-dimension of the model class $\mathcal{F}$.[1]

**Theorem 3.** *Assume that the cross-entropy loss or MSE loss is used to instantiate $\mathcal{L}$. The following inequality holds with probability at least $1 - \delta$:*

$$\tilde{R}(\hat{f}_1, \hat{f}_2, \ldots, \hat{f}_q) - \tilde{R}(f_1^*, f_2^*, \ldots, f_q^*) \leqslant \bar{\mathcal{O}}_p\left(q\sqrt{\frac{dm^3}{n}}\right), \tag{17}$$

*where $\bar{\mathcal{O}}_p$ hides logarithmic factors $\log m$ and $\log n$, and the subscript $p$ denotes the order in probability.*

The proof is given in Appendix B.5. Theorem 3 shows an estimation error bound for the proposed risk estimator. When the number of bags tends to infinity, the risk of the classifiers obtained by minimizing Eq. (9) converges to the risk of the optimal classifiers. The former works (Busa-Fekete et al., 2023; 2025) differ from ours in two respects. First, they impose a stronger within-bag i.i.d. assumption on instances. Under this stronger assumption, they can recover instance-level posterior probabilities, whereas under our weaker assumption we can only recover bag-level posterior probabilities. Consequently, their bounds are not directly comparable with ours.

## 3.4 AN EFFICIENT COMPUTATIONAL SOLUTION

Computing $g_{k,l}(\mathbf{B})$ is equivalent to computing the coefficients $c_{k,l-1}$ in Eq. (16), which involves the product of $m$ binomials. The time complexity of computing Eq. (16) is $2^m$, which is intractable for large bag sizes. To solve this problem, we propose a novel, significantly more efficient method that leverages the fast Fourier transform (FFT).[2] The *Convolution Theorem* (Oppenheim, 1999; Cormen et al., 2022) states that the coefficients of the product of two polynomials can be obtained by taking the inverse Discrete Fourier Transform (DFT) of the pointwise product of their DFT-transformed coefficient vectors. Since the time complexities of the product computation can be greatly reduced in the frequency domain, this inspires us to use the DFT to compute Eq. (16). We use a *divide-and-conquer* approach by calculating the product of polynomials pairwise with multiple iterations of a repetitive procedure. First, we group the $m$ binomials pairwise in sequence and pad the coefficient vector for each binomial as

$$\boldsymbol{h}_{k,j}^0 = \left[h_{k,j,0}^0, h_{k,j,1}^0, \ldots, h_{k,j,m}^0\right] = [1 - f_k(\boldsymbol{x}_j), \ f_k(\boldsymbol{x}_j), \ 0, \ \ldots, \ 0] \in \mathbb{R}^{m+1},$$

where $h_{k,j,t}^0$ denotes the $t$-th order coefficient for the $j$-th binomial. Then we transform $\boldsymbol{h}_{k,j}^0$ into its frequency-domain representation $\boldsymbol{H}_{k,j}^0 = \left[H_{k,j,0}^0, H_{k,j,1}^0, \ldots, H_{k,j,m}^0\right]$ by the $(m+1)$-point DFT. Specifically, we have

$$H_{k,j,t}^0 = \sum_{t'=0}^m h_{k,j,t'}^0 \ \exp\left(-2\pi i \frac{tt'}{m+1}\right), \quad t \in \{0, 1, 2, \ldots, m\}. \tag{18}$$

Then, in the $s$-th iteration, the goal is to compute the product of $N_s$ polynomials grouped pairwise, and we have $N_1 = m$. By the Convolution Theorem, the frequency representation of the coefficients for the polynomial of the product in each group can be determined as the elementwise product of the frequency representations of each polynomial, i.e.,

$$\boldsymbol{H}_{k,j}^s = \begin{cases} \boldsymbol{H}_{k,2j-1}^{s-1} \odot \boldsymbol{H}_{k,2j}^{s-1}, & j \in \left\{1, 2, \ldots, \left\lfloor \frac{N_s}{2} \right\rfloor\right\}; \\ \boldsymbol{H}_{k,2j-1}^{s-1}, & j = \frac{N_s+1}{2}, \text{ and } N_s \text{ is odd}, \end{cases} \tag{19}$$

---

[1]Pseudo-dimension extends VC dimension to real-valued classes (Anthony & Bartlett, 2009).
[2]https://github.com/pytorch/pytorch/blob/main/torch/fft

where $\odot$ denotes elementwise product and $\lfloor \cdot \rfloor$ denotes rounding down to the nearest integer. Here, when $N_s$ is an odd number, we leave the last polynomial unchanged for the next iteration. Then, $\left\{ \boldsymbol{H}_{k,j}^s \right\}$ is used as the frequency-domain representation of the polynomials for the product in the $(s+1)$-th iteration. After $\lceil \log_2(m+1) \rceil$ iterations, we arrive at the target single frequency-domain representation $\boldsymbol{H}_k^* = \boldsymbol{H}_{k,1}^{\lceil \log_2(m+1) \rceil}$, where $\lceil \cdot \rceil$ denotes rounding up to the nearest integer. Notably, $\boldsymbol{H}_k^*$ is the frequency-domain representation of the coefficients of the polynomial in Eq. (16). Then, we recover the coefficients from $\boldsymbol{H}_k^*$ by applying the $(m+1)$-point inverse DFT. Specifically, for $r \in \{0, 1, \ldots, m\}$, we have

$$c_{k,r} = \frac{1}{m+1} \sum_{t=0}^{m} H_{k,t}^* \exp\left( +2\pi i \, \frac{r\,t}{m+1} \right). \tag{20}$$

Finally, according to Lemma 2, we have $g_{k,l}(\mathbf{B}) = c_{k,l-1}$.

**Proposition 1.** *For a bag* $\mathbf{B}$ *of size* $m$, *computing the model output* $g_{k,l}(\mathbf{B})$ *via the divide-and-conquer-based algorithm on a graphics processing unit (GPU) has a runtime of* $O(m \log m)$.

The proof is given in Appendix B.6. Compared with the dynamic programming-based method (Shukla et al., 2023) with a time complexity of $O(m^2)$, our lower time complexity benefits from the efficient execution of operations in Eqs. (18), (19), and (20) on GPUs.

## 4 EXPERIMENTS

In this section, we first present the experimental setup. Then, we present the experimental results and the execution time analysis.

### 4.1 EXPERIMENTAL SETUP

We used four widely used datasets, including Kuzushiji-MNIST (K-MNIST), Fashion-MNIST (F-MNIST), SVHN, and CIFAR-10. We adopted three different bag generation strategies: i) Random bag: the dataset is randomly shuffled and then randomly assembled into bags; ii) Cluster bag 3: we first cluster the dataset into a predefined number of clusters, and then form bags by sampling instances from these cluster-based groups; iii) $\boldsymbol{\alpha}$-First bag 4: for each bag, we first sample a bag-level class proportion from a Dirichlet distribution, and then draw instances from each class pool to match the sampled proportions. For all bag generation strategies, the bag size $m$ is varied as powers of two, i.e., $m \in \{4, 8, \ldots, 128\}$.

We compared our approach against five state-of-the-art LLP methods: PM (Yu et al., 2014; Ardehaly & Culotta, 2017), DSQ (Li et al., 2024a), EasyLLP (Busa-Fekete et al., 2023), EasyLLP-flood, and ROT (Dulac-Arnold et al., 2019). Here, EasyLLP-flood refers to EasyLLP enhanced with the flooding regularization technique (Ishida et al., 2020). To ensure fair comparisons, all methods shared the same hyperparameter settings. More details can be found in Appendix D.

### 4.2 EXPERIMENTAL RESULTS

Tables 1, 2, and 3 present the experimental results for various datasets and bag generation strategies, showing that LLP-PVC consistently achieves state-of-the-art performance across different bag sizes. First, LLP-PVC can recover an optimal set of instance-level predictions; together with the representation learning capability of neural networks, the model can further assign these predictions to the correct instances across bags. Second, LLP-PVC learns more discriminative instance-level decision functions, rather than merely fitting bag-level label proportions, as standard proportion-matching approaches do. Third, our formulation is straightforward to optimize in practice and avoids known optimization pitfalls of some prior methods (e.g., the negative-risk issue in EasyLLP), leading to stable training and strong performance. Fourth, our method does not rely on the within-bag i.i.d. assumption and remains robust across diverse bag-generation mechanisms.

### 4.3 EXECUTION TIME ANALYSIS

Because computing the loss requires an extra processing step, we report the corresponding runtime in Table 4. We additionally evaluate two methods, UUM (Wei et al., 2023) and Count

Table 1: Accuracy results (mean ± std) under **Random Bag** construction. The best-performing method for each bag size is highlighted in bold.

| Dataset | Method | 4 | 8 | 16 | 32 | 64 | 128 |
|---|---|---|---|---|---|---|---|
| K-MNIST | PM | 93.19 ± 0.99 | 90.88 ± 0.87 | 83.52 ± 1.01 | 71.10 ± 2.07 | 51.05 ± 4.22 | 28.30 ± 1.29 |
| | DSQ | 59.15 ± 0.85 | 59.28 ± 0.73 | 58.87 ± 2.26 | 57.74 ± 1.21 | 57.58 ± 2.52 | 54.90 ± 2.41 |
| | EasyLLP | 72.56 ± 2.32 | 67.40 ± 4.28 | 63.48 ± 5.33 | 56.93 ± 1.36 | 50.60 ± 1.46 | 48.56 ± 2.84 |
| | EasyLLP-flood | 86.47 ± 0.21 | 83.89 ± 0.99 | 81.03 ± 0.56 | 75.99 ± 2.12 | 70.48 ± 2.13 | 63.87 ± 1.26 |
| | ROT | 96.40 ± 0.41 | 96.21 ± 0.24 | 94.29 ± 0.31 | 90.14 ± 0.51 | 73.00 ± 4.23 | 43.37 ± 2.37 |
| | LLP-PVC | **96.55 ± 0.09** | **96.62 ± 0.29** | **96.68 ± 0.22** | **96.10 ± 0.15** | **96.76 ± 0.15** | **96.36 ± 0.16** |
| F-MNIST | PM | 88.36 ± 0.57 | 86.87 ± 0.55 | 84.44 ± 0.73 | 80.21 ± 1.08 | 75.81 ± 0.68 | 66.78 ± 5.29 |
| | DSQ | 77.82 ± 0.76 | 77.91 ± 0.43 | 77.73 ± 0.77 | 77.22 ± 0.80 | 77.14 ± 0.42 | 75.82 ± 0.82 |
| | EasyLLP | 74.31 ± 2.22 | 73.59 ± 1.83 | 72.27 ± 4.71 | 71.75 ± 1.57 | 68.13 ± 0.39 | 66.27 ± 0.84 |
| | EasyLLP-flood | 85.89 ± 0.11 | 85.89 ± 0.29 | 83.45 ± 0.18 | 82.06 ± 0.85 | 79.22 ± 0.37 | 73.69 ± 0.35 |
| | ROT | **91.43 ± 0.27** | 91.02 ± 0.48 | 88.87 ± 0.60 | 86.75 ± 0.68 | 81.42 ± 0.34 | 65.38 ± 5.20 |
| | LLP-PVC | 91.39 ± 0.33 | **91.84 ± 0.15** | **91.70 ± 0.28** | **90.32 ± 0.42** | **89.28 ± 0.02** | **87.17 ± 0.16** |
| SVHN | PM | 92.50 ± 0.10 | 91.36 ± 0.07 | 88.57 ± 0.21 | 81.22 ± 0.34 | 64.23 ± 0.77 | 43.18 ± 0.72 |
| | DSQ | 78.22 ± 0.12 | 77.77 ± 0.23 | 76.61 ± 0.07 | 73.97 ± 0.65 | 65.79 ± 1.37 | 50.03 ± 1.50 |
| | EasyLLP | 76.95 ± 1.27 | 68.87 ± 1.10 | 59.77 ± 1.10 | 51.49 ± 1.02 | 37.76 ± 2.31 | 32.62 ± 2.18 |
| | EasyLLP-flood | 84.90 ± 0.44 | 81.06 ± 0.55 | 75.28 ± 0.62 | 67.77 ± 0.49 | 57.97 ± 0.81 | 47.80 ± 3.16 |
| | ROT | 92.98 ± 0.07 | 92.22 ± 0.13 | 90.43 ± 0.11 | 85.64 ± 0.30 | 71.91 ± 0.64 | 50.88 ± 1.45 |
| | LLP-PVC | **94.30 ± 0.05** | **94.50 ± 0.06** | **94.56 ± 0.14** | **94.61 ± 0.08** | **94.27 ± 0.15** | **93.60 ± 0.76** |
| CIFAR-10 | PM | 82.03 ± 0.34 | 80.23 ± 0.51 | 74.74 ± 0.42 | 68.51 ± 0.15 | 60.47 ± 0.20 | 49.21 ± 0.38 |
| | DSQ | 69.50 ± 0.43 | 69.32 ± 0.27 | 67.99 ± 0.50 | 65.74 ± 0.22 | 60.70 ± 0.16 | 51.35 ± 0.64 |
| | EasyLLP | 66.89 ± 0.18 | 63.47 ± 0.59 | 56.92 ± 0.75 | 51.81 ± 0.52 | 44.19 ± 0.94 | 37.36 ± 1.87 |
| | EasyLLP-flood | 69.24 ± 0.19 | 66.62 ± 0.12 | 62.28 ± 0.37 | 58.86 ± 0.45 | 54.88 ± 0.44 | 47.54 ± 0.41 |
| | ROT | 83.76 ± 0.25 | 82.02 ± 0.18 | 80.37 ± 0.09 | 77.09 ± 0.07 | 69.59 ± 1.28 | 33.70 ± 8.02 |
| | LLP-PVC | **84.34 ± 0.26** | **84.43 ± 0.33** | **84.50 ± 0.22** | **82.44 ± 0.29** | **78.90 ± 0.44** | **76.01 ± 0.97** |

Table 2: Accuracy results (mean ± std over seeds) under **Cluster Bag** construction. The best method for each bag size is in bold.

| Dataset | Method | 4 | 8 | 16 | 32 | 64 | 128 |
|---|---|---|---|---|---|---|---|
| K-MNIST | PM | 92.89 ± 0.15 | 91.05 ± 0.41 | 86.94 ± 0.11 | 81.60 ± 1.11 | 76.56 ± 1.25 | 72.49 ± 1.61 |
| | DSQ | 67.92 ± 3.03 | 72.32 ± 0.79 | 74.57 ± 1.04 | 75.19 ± 1.06 | 75.56 ± 2.12 | 75.24 ± 2.31 |
| | EasyLLP | 31.61 ± 0.66 | 21.05 ± 1.25 | 17.22 ± 3.41 | 14.92 ± 6.98 | 9.98 ± 0.00 | 9.98 ± 0.00 |
| | EasyLLP-flood | 60.95 ± 0.68 | 57.08 ± 0.01 | 50.42 ± 1.13 | 41.85 ± 2.96 | 35.27 ± 2.77 | 26.72 ± 0.63 |
| | ROT | 94.64 ± 0.19 | 93.68 ± 0.09 | 91.62 ± 0.33 | 87.35 ± 0.25 | 82.51 ± 1.22 | 77.04 ± 0.95 |
| | LLP-PVC | **96.04 ± 0.14** | **96.47 ± 0.17** | **96.57 ± 0.19** | **96.66 ± 0.11** | **96.63 ± 0.17** | **96.11 ± 0.19** |
| F-MNIST | PM | 88.25 ± 0.30 | 86.75 ± 0.31 | 84.44 ± 0.12 | 81.88 ± 0.48 | 79.23 ± 1.02 | 77.60 ± 0.41 |
| | DSQ | 78.99 ± 0.16 | 79.38 ± 0.33 | 79.61 ± 0.52 | 79.44 ± 0.25 | 78.96 ± 0.52 | 78.76 ± 0.76 |
| | EasyLLP | 26.82 ± 1.40 | 16.24 ± 4.76 | 12.64 ± 3.68 | 10.04 ± 0.00 | 10.04 ± 0.00 | 10.04 ± 0.00 |
| | EasyLLP-flood | 73.07 ± 0.62 | 71.33 ± 0.03 | 63.54 ± 0.78 | 53.03 ± 3.80 | 46.88 ± 1.55 | 40.91 ± 2.68 |
| | ROT | 89.93 ± 0.26 | 88.48 ± 0.20 | 86.77 ± 0.10 | 84.18 ± 0.46 | 81.62 ± 1.13 | 79.41 ± 0.63 |
| | LLP-PVC | **91.47 ± 0.06** | **91.68 ± 0.06** | **91.39 ± 0.20** | **90.51 ± 0.12** | **89.16 ± 0.23** | **87.15 ± 0.29** |
| SVHN | PM | 92.50 ± 0.09 | 91.34 ± 0.06 | 88.75 ± 0.08 | 81.93 ± 0.31 | 66.84 ± 1.25 | 48.69 ± 1.24 |
| | DSQ | 78.34 ± 0.12 | 77.82 ± 0.15 | 76.90 ± 0.13 | 74.57 ± 0.17 | 68.57 ± 0.22 | 57.05 ± 1.83 |
| | EasyLLP | 70.54 ± 0.71 | 53.57 ± 0.48 | 36.18 ± 0.95 | 27.16 ± 0.75 | 20.62 ± 1.07 | 19.86 ± 0.48 |
| | EasyLLP-flood | 84.86 ± 0.03 | 81.32 ± 0.14 | 75.84 ± 0.53 | 67.29 ± 0.61 | 60.04 ± 0.10 | 48.79 ± 4.00 |
| | ROT | 92.90 ± 0.03 | 92.26 ± 0.08 | 90.38 ± 0.09 | 86.07 ± 0.15 | 74.83 ± 0.72 | 58.33 ± 1.17 |
| | LLP-PVC | **94.34 ± 0.09** | **94.43 ± 0.15** | **94.61 ± 0.15** | **94.84 ± 0.23** | **94.54 ± 0.02** | **93.89 ± 0.65** |
| CIFAR-10 | PM | 81.33 ± 0.28 | 79.06 ± 0.60 | 75.22 ± 0.14 | 69.66 ± 0.52 | 64.00 ± 0.59 | 58.67 ± 0.53 |
| | DSQ | 69.97 ± 0.33 | 69.59 ± 0.21 | 68.64 ± 0.35 | 66.35 ± 0.51 | 63.28 ± 0.47 | 59.55 ± 0.45 |
| | EasyLLP | 55.32 ± 0.26 | 43.72 ± 0.91 | 33.71 ± 0.61 | 24.92 ± 0.97 | 19.57 ± 2.38 | 16.46 ± 0.33 |
| | EasyLLP-flood | 67.78 ± 0.51 | 61.08 ± 0.53 | 52.67 ± 1.02 | 42.12 ± 1.93 | 34.10 ± 0.99 | 29.41 ± 0.45 |
| | ROT | 82.11 ± 0.57 | 80.31 ± 0.17 | 76.97 ± 0.13 | 71.42 ± 0.51 | 65.94 ± 0.25 | 61.14 ± 0.92 |
| | LLP-PVC | **84.63 ± 0.09** | **84.61 ± 0.06** | **83.97 ± 0.04** | **82.00 ± 0.38** | **79.94 ± 0.19** | **78.36 ± 0.50** |

Loss (Shukla et al., 2023). For Count Loss, we directly extend the original count-based formulation to the multiclass classification setting. As shown in Table 4, our method achieves nearly the same runtime as $O(1)$-time baselines because the loss can be computed entirely on the GPU with full parallelism. Due to the high time complexity, UUM and Count-Loss are not suitable for large bags, limiting their practical applicability. As shown in Figure 2, the execution time of LLP-PVC remains consistently low as the bag size increases, while LLP-PVC (DP) increases significantly with larger bag sizes, becoming over ten times slower when the bag size is large.

Table 3: Accuracy results (mean $\pm$ std over seeds) under $\boldsymbol{\alpha}$**-First Bag** construction. The best method for each bag size is in bold.

| Dataset | Method | 4 | 8 | 16 | 32 | 64 | 128 |
|---------|--------|---|---|----|----|----|-----|
| K-MNIST | PM | $93.07 \pm 0.28$ | $90.25 \pm 0.82$ | $71.58 \pm 1.62$ | $43.23 \pm 2.39$ | $37.03 \pm 2.77$ | $33.16 \pm 3.89$ |
| | DSQ | $58.63 \pm 3.11$ | $52.45 \pm 3.39$ | $32.15 \pm 2.91$ | $29.26 \pm 2.60$ | $40.39 \pm 5.71$ | $59.19 \pm 1.33$ |
| | EasyLLP | $63.66 \pm 1.21$ | $78.26 \pm 1.72$ | $74.43 \pm 1.52$ | $67.08 \pm 1.67$ | $60.56 \pm 2.87$ | $36.78 \pm 2.15$ |
| | EasyLLP-flood | $82.58 \pm 0.20$ | $81.07 \pm 1.26$ | $74.59 \pm 2.33$ | $66.92 \pm 1.83$ | $62.28 \pm 1.65$ | $64.41 \pm 2.03$ |
| | ROT | $94.84 \pm 0.18$ | $93.58 \pm 0.32$ | $82.85 \pm 2.00$ | $68.31 \pm 3.27$ | $59.21 \pm 4.17$ | $54.59 \pm 2.10$ |
| | LLP-PVC | $\mathbf{95.29 \pm 1.26}$ | $\mathbf{96.02 \pm 0.37}$ | $\mathbf{96.50 \pm 0.43}$ | $\mathbf{96.42 \pm 0.14}$ | $\mathbf{96.61 \pm 0.13}$ | $\mathbf{96.03 \pm 0.11}$ |
| F-MNIST | PM | $88.59 \pm 0.18$ | $86.66 \pm 0.09$ | $79.50 \pm 0.62$ | $75.34 \pm 0.33$ | $72.46 \pm 0.71$ | $70.02 \pm 0.89$ |
| | DSQ | $78.00 \pm 0.31$ | $76.91 \pm 0.27$ | $71.21 \pm 1.70$ | $71.28 \pm 2.23$ | $75.33 \pm 0.20$ | $76.76 \pm 0.49$ |
| | EasyLLP | $69.23 \pm 0.97$ | $81.83 \pm 0.62$ | $81.64 \pm 0.23$ | $78.80 \pm 0.06$ | $74.09 \pm 0.74$ | $45.69 \pm 6.23$ |
| | EasyLLP-flood | $85.40 \pm 0.06$ | $84.11 \pm 0.53$ | $81.94 \pm 0.61$ | $78.54 \pm 0.35$ | $75.09 \pm 0.22$ | $77.49 \pm 0.58$ |
| | ROT | $90.01 \pm 0.10$ | $88.61 \pm 0.19$ | $83.51 \pm 0.68$ | $78.93 \pm 0.66$ | $76.44 \pm 0.36$ | $74.36 \pm 0.34$ |
| | LLP-PVC | $\mathbf{90.52 \pm 1.96}$ | $\mathbf{91.19 \pm 0.94}$ | $\mathbf{91.53 \pm 0.22}$ | $\mathbf{90.56 \pm 0.18}$ | $\mathbf{89.32 \pm 0.71}$ | $\mathbf{87.55 \pm 0.34}$ |
| SVHN | PM | $92.46 \pm 0.05$ | $91.32 \pm 0.02$ | $83.85 \pm 0.06$ | $73.43 \pm 0.07$ | $67.02 \pm 0.18$ | $61.44 \pm 0.21$ |
| | DSQ | $79.52 \pm 0.07$ | $75.99 \pm 0.15$ | $63.44 \pm 0.20$ | $63.47 \pm 0.21$ | $67.63 \pm 0.14$ | $66.93 \pm 0.09$ |
| | EasyLLP | $76.77 \pm 0.41$ | $67.45 \pm 0.66$ | $45.81 \pm 0.84$ | $39.43 \pm 1.02$ | $35.84 \pm 1.06$ | $31.98 \pm 2.61$ |
| | EasyLLP-flood | $86.48 \pm 0.14$ | $79.13 \pm 0.19$ | $62.88 \pm 0.70$ | $58.08 \pm 2.27$ | $63.69 \pm 0.80$ | $61.00 \pm 0.80$ |
| | ROT | $92.93 \pm 0.10$ | $92.09 \pm 0.09$ | $87.98 \pm 0.21$ | $80.93 \pm 0.19$ | $75.18 \pm 0.30$ | $69.81 \pm 0.23$ |
| | LLP-PVC | $\mathbf{94.35 \pm 0.02}$ | $\mathbf{94.45 \pm 0.08}$ | $\mathbf{94.55 \pm 0.03}$ | $\mathbf{94.53 \pm 0.11}$ | $\mathbf{94.31 \pm 0.09}$ | $\mathbf{94.20 \pm 0.28}$ |
| CIFAR-10 | PM | $81.42 \pm 0.19$ | $78.89 \pm 0.14$ | $64.46 \pm 0.39$ | $56.77 \pm 0.65$ | $54.08 \pm 0.90$ | $54.65 \pm 0.92$ |
| | DSQ | $70.13 \pm 0.20$ | $67.28 \pm 0.19$ | $55.60 \pm 0.62$ | $53.76 \pm 0.91$ | $54.64 \pm 0.62$ | $56.33 \pm 0.91$ |
| | EasyLLP | $67.22 \pm 0.13$ | $61.02 \pm 0.16$ | $46.23 \pm 0.61$ | $41.42 \pm 0.77$ | $41.06 \pm 0.59$ | $40.71 \pm 0.94$ |
| | EasyLLP-flood | $70.85 \pm 0.12$ | $61.80 \pm 0.34$ | $46.88 \pm 0.57$ | $41.69 \pm 0.26$ | $48.28 \pm 0.44$ | $54.20 \pm 0.36$ |
| | ROT | $82.42 \pm 0.29$ | $80.32 \pm 0.19$ | $66.43 \pm 0.16$ | $59.36 \pm 0.49$ | $56.22 \pm 0.55$ | $57.28 \pm 0.69$ |
| | LLP-PVC | $\mathbf{84.89 \pm 0.19}$ | $\mathbf{84.83 \pm 0.13}$ | $\mathbf{83.08 \pm 0.22}$ | $\mathbf{80.18 \pm 0.09}$ | $\mathbf{76.70 \pm 0.20}$ | $\mathbf{76.36 \pm 0.34}$ |

Table 4: Execution time results (mean over 30 epochs) under the same settings on a single NVIDIA A100 GPU. "GPU parallelizable" indicates whether bags within a batch can be processed in parallel on the GPU. "–" indicates that results could not be obtained within a reasonable time due to the combinatorial explosion of bag size. $t$ denotes the number of iterations used when computing the optimal transport (e.g., in the Sinkhorn algorithm).

| Method | K-MNIST | | | | CIFAR-10 | | | | Time Complexity (per bag) | GPU Parallelizable |
|--------|---------|---|----|----|----------|---|----|----|---------|-----|
| | 4 | 8 | 16 | 32 | 4 | 8 | 16 | 32 | | |
| PM | 2.59 | 2.63 | 2.45 | 2.49 | 3.63 | 3.46 | 3.86 | 2.97 | $O(1)$ | ✓ |
| DSQ | 2.53 | 2.72 | 2.04 | 2.08 | 3.28 | 3.67 | 3.59 | 2.88 | $O(1)$ | ✓ |
| EasyLLP | 2.89 | 2.87 | 1.81 | 1.62 | 3.07 | 3.01 | 2.91 | 2.66 | $O(1)$ | ✓ |
| EasyLLP-flood | 2.92 | 2.69 | 1.77 | 1.67 | 3.13 | 2.99 | 2.78 | 2.33 | $O(1)$ | ✓ |
| ROT | 20.92 | 11.63 | 5.85 | 3.56 | 17.83 | 10.82 | 6.68 | 4.55 | $O(q\,m\,t)$ | ✗ |
| UUM | 35.88 | 55.62 | – | – | 28.37 | 44.25 | – | – | $O(m!)$ | ✗ |
| Count Loss | 119.23 | 172.34 | – | – | 98.01 | 139.52 | – | – | $O(m!)$ | ✗ |
| LLP-PVC | 2.77 | 2.91 | 1.78 | 1.80 | 3.02 | 3.01 | 3.48 | 2.93 | $O(m \log m)$ | ✓ |

## 5 CONCLUSION

In this paper, we proposed a novel approach for LLP by solving an auxiliary proportional value classification problem. We demonstrated that our approach can mitigate the over-smoothing issues of the proportion matching strategies both theoretically and empirically. We also introduced an efficient computational approach with a divide-and-conquer strategy. Extensive experiments with different bag construction strategies validated the effectiveness of the proposed approach. In future work, we aim to extend our framework to such dynamic environments, guided by novel metrics like OpenworldAUC (Hua et al., 2025), and to incorporate differential privacy (Dwork et al., 2006) to protect instance-level data.

ACKNOWLEDGMENTS

WW was supported by the Junior Research Associate (JRA) program of RIKEN. MS was supported by JST ASPIRE Grant Number JPMJAP2405.

ETHICS STATEMENT

This paper presents work that focuses on the methodology of weakly supervised machine learning and does not raise any ethical concerns.

REPRODUCIBILITY STATEMENT

The details of the experimental setup can be found in Appendix D. The code implementation can be found at https://github.com/TianhaoMa5/ICLR2026_LLP-PVC.

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

# A  RELATED WORK

To train an instance-level classifier from unlabeled bags with label proportions, several related works build on the proportion-matching loss and enhance it with additional regularization components, such as consistency regularization (Tsai & Lin, 2020; Yang et al., 2021), generative adversarial networks (Liu et al., 2019), and pseudo-labeling (Liu et al., 2021; Ma et al., 2025). Notably, these components are widely used across weakly supervised learning in general, often independently of the LLP setting.

A large body of existing methods assumes that instances within each bag are i.i.d., and under this assumption, a variety of algorithms with theoretical guarantees have been developed. Wei et al. (2023) proposed UUM, a risk-consistent approach using the importance weighting technique. Although UUM has solid theoretical guarantees, the computational complexity of the posterior probabilities grows exponentially as the number of instances in each bag increases. EasyLLP (Busa-Fekete et al., 2023) and GeneralUPM (Applebaum et al., 2025) both derive an *unbiased risk estimator* for the instance-level classification risk, yielding a method that is simple to implement yet theoretically well-founded. However, the potentially negative terms in the risk estimator can often lead to severe overfitting problems and performance degeneration (Li et al., 2024b), which is a common phenomenon in other weakly supervised learning problems (Kiryo et al., 2017; Lu et al., 2020; Sugiyama et al., 2022). Li et al. (2024b) proposed DSQ, a variant of the proportion matching strategy. DSQ enjoys the optimistic convergence rate under both feasible and agnostic cases. However, DSQ restricts the loss function to the mean squared loss, which may be suboptimal in many classification cases. The latest study (Busa-Fekete et al., 2025) considers only the binary setting and is developed specifically for the mean squared loss. The count loss (Shukla et al., 2023) does not rely on the within-bag instance i.i.d. assumption. It adopts a maximum-likelihood formulation and provides a dynamic-programming-based procedure to compute the posterior probability of the label proportion within a bag. However, it is formulated for binary classification, and a straightforward extension to the multiclass setting would incur prohibitively high computational cost. These drawbacks make them less useful in real-world LLP applications where requirements are difficult to meet. Table 5 summarizes the characteristics of different state-of-the-art LLP approaches and our proposed approach.

In addition, we adopt the traditional LLP setting where bags are sampled i.i.d., while allowing the instances within each bag to be potentially non-i.i.d. This is arguably more practical, since in real-world applications bags are often constructed by heuristic grouping (Chen et al., 2023; Tsai & Lin, 2020), which can induce within-bag correlations. Simultaneously, we do not include approaches from related but different settings for comparison, since they are often tailored to more specialized assumptions or problem formulations. Unlabeled-unlabeled (UU) learning (Lu et al., 2019; 2022; Wei et al., 2024) considered the class priors of different bags to be different and we do not include related approaches for comparison (Scott & Zhang, 2020; Zhang et al., 2022). Saket (2021; 2022) allow the same instances to appear in multiple bags simultaneously. Brahmbhatt et al. (2023) further assume that instances are sampled from a Gaussian distribution.

Table 5: Comparison of different LLP approaches.

| Approach | Dealing with 2+ classes | No negative loss risk | Compatible with large bag sizes | Compatible with any loss function | No i.i.d. assumption about within-bag instances |
|---|---|---|---|---|---|
| PM (Yu et al., 2014) | ✓ | ✓ | ✓ | ✓ | ✓ |
| UUM (Wei et al., 2023) | ✓ | ✓ | ✗ | ✓ | ✗ |
| EasyLLP (Busa-Fekete et al., 2023) | ✓ | ✗ | ✓ | ✓ | ✗ |
| Count Loss (Shukla et al., 2023) | ✓ | ✓ | ✗ | ✓ | ✓ |
| DSQ (Li et al., 2024b) | ✓ | ✓ | ✓ | ✗ | ✗ |
| Busa-Fekete et al. (2025) | ✗ | ✓ | ✓ | ✗ | ✗ |
| GeneralUPM (Applebaum et al., 2025) | ✓ | ✗ | ✓ | ✓ | ✗ |
| LLP-PVC | ✓ | ✓ | ✓ | ✓ | ✓ |

# B PROOF

## B.1 PROOF OF LEMMA 1

$$
\begin{aligned}
p(\tilde{\alpha}_k = l | \mathbf{B}) &= p\left(\sum_{j=1}^m \tilde{y}_j^k = l | \mathbf{B}\right) \\
&= \sum_{r=1}^{2^m} \mathbb{I}\left(\mathrm{HW}\left(\mathrm{Bin}(r-1,m)\right) + 1 = l\right) \prod_{j=1}^m p\left(\tilde{y}_j^k = b_j^{(r-1)} | \boldsymbol{x}_j\right) \\
&= \sum_{r=1}^{2^m} q_{r,l} \prod_{j=1}^m \left(p\left(\tilde{y}_j^k = 1 | \boldsymbol{x}_j\right)^{b_j^{(r-1)}} p\left(\tilde{y}_j^k = 0 | \boldsymbol{x}_j\right)^{\left(1 - b_j^{(r-1)}\right)}\right),
\end{aligned}
$$

which concludes the proof. $\qquad\square$

## B.2 PROOF OF LEMMA 2

For each $r \in \{1, \dots, 2^m\}$, we have

$$
\mathrm{Bin}(r-1, m) = \left(b_1^{(r-1)}, \dots, b_m^{(r-1)}\right), \qquad S_r = \{\, j \mid b_j^{(r-1)} = 1 \,\} \subseteq [m].
$$

It is obvious that $|S_r| = \sum_{j=1}^m b_j^{(r-1)}$. Since $q_{r,l} = p(\tilde{\alpha}_k = l | \tilde{\boldsymbol{y}}^k = \mathrm{Bin}(r-1, m)) = \mathbb{I}(|S_r| = l-1)$, we have

$$
\begin{aligned}
g_{k,l}(\mathbf{B}) &= \sum_{r=1}^{2^m} q_{r,l} \prod_{j=1}^m f_k(\boldsymbol{x}_j)^{b_j^{(r-1)}} \left(1 - f_k(\boldsymbol{x}_j)\right)^{1 - b_j^{(r-1)}} \\
&= \sum_{r=1}^{2^m} \mathbb{I}(|S_r| = l-1) \prod_{j \in S_r} f_k(\boldsymbol{x}_j) \prod_{j \notin S_r} \left(1 - f_k(\boldsymbol{x}_j)\right) \\
&= \sum_{\substack{S \subseteq [m] \\ |S| = l-1}} \prod_{j \in S} f_k(\boldsymbol{x}_j) \prod_{j \notin S} \left(1 - f_k(\boldsymbol{x}_j)\right).
\end{aligned}
\tag{21}
$$

We use the elementary identity

$$
\prod_{j=1}^m (a_j + b_j) = \sum_{S \subseteq [m]} \left(\prod_{j \in S} a_j\right)\left(\prod_{j \notin S} b_j\right).
$$

Taking $a_j = f_k(\boldsymbol{x}_j)\, z$ and $b_j = 1 - f_k(\boldsymbol{x}_j)$, we have

$$
\begin{aligned}
G_k(z) &= \sum_{S \subseteq [m]} \left(\prod_{j \in S} f_k(\boldsymbol{x}_j)\right)\left(\prod_{j \notin S} \left(1 - f_k(\boldsymbol{x}_j)\right)\right) z^{|S|} \\
&= \sum_{t=0}^m c_{k,t}\, z^t,
\end{aligned}
\tag{22}
$$

where

$$
c_{k,t} = \sum_{\substack{S \subseteq [m] \\ |S| = t}} \left(\prod_{j \in S} f_k(\boldsymbol{x}_j)\right)\left(\prod_{j \notin S} \left(1 - f_k(\boldsymbol{x}_j)\right)\right).
\tag{23}
$$

Based on (21) and (23), setting $t = l - 1$ shows that

$$
c_{k,t} = g_{k,t+1}(\mathbf{B}), \qquad t \in \{0, \dots, m\}.
$$

The proof is completed. $\qquad\square$

## B.3 PROOF OF THEOREM 1

Firstly, we introduce a binomial for $f_k(\boldsymbol{x}_j), j \in \{1, \dots, m\}$:

$$
F_{k,j}(z) = f_k(\boldsymbol{x}_j) z + \left(1 - f_k(\boldsymbol{x}_j)\right).
$$

Then, we define an $m$-order polynomial as the product of all $m$ binomials:

$$
G_k(z) = \prod_{j=1}^m F_{k,j}(z) = \prod_{j=1}^m \left(f_k(\boldsymbol{x}_j) z + (1 - f_k(\boldsymbol{x}_j))\right) = \sum_{t=0}^m c_{k,t}\, z^t,
$$

where $c_{k,t}$ is the coefficient of $z^t$, which can be determined by Lemma 2. The $m$-order polynomial $G_k(z)$ have $m$ roots $\{1 - 1/f_k(\boldsymbol{x}_j)\}_{j=1}^m$. Suppose there exist another different set of model outputs

of the instance-level classifier $\{f'_k(\boldsymbol{x}_j)\}_{j=1}^m$ that can lead to the same model output of the bag-level classifier, i.e.

$$\forall l \in \{1, 2, \ldots, m+1\}, \; g_{k,l}(\mathbf{B}) = \sum_{r=1}^{2^m} q_{r,l} \prod_{j=1}^m \left( f_k(\boldsymbol{x}_j)^{b_j^{(r-1)}} \left(1 - f_k(\boldsymbol{x}_j)\right)^{\left(1 - b_j^{(r-1)}\right)} \right)$$

$$= \sum_{r=1}^{2^m} q_{r,l} \prod_{j=1}^m \left( f'_k(\boldsymbol{x}_j)^{b_j^{(r-1)}} \left(1 - f'_k(\boldsymbol{x}_j)\right)^{\left(1 - b_j^{(r-1)}\right)} \right).$$

We also introduce a binomial for $f'_k(\boldsymbol{x}_j), j \in \{1, \ldots, m\}$:

$$F'_{k,j}(z) = f'_k(\boldsymbol{x}_j)z + \left(1 - f'_k(\boldsymbol{x}_j)\right).$$

We also define an $m$-order polynomial as the product of all $m$ binomials:

$$G'_k(z) = \prod_{j=1}^m F'_{k,j}(z) = \prod_{j=1}^m \left( f'_k(\boldsymbol{x}_j)z + (1 - f'_k(\boldsymbol{x}_j)) \right) = \sum_{t=0}^m c'_{k,t} z^t,$$

We have $c_{k,t} = c'_{k,t}$ by Lemma 2. Therefore, the roots of $G_k(z)$ are the same as $G'_k(z)$, i.e.,

$$\{1 - 1/f_k(\boldsymbol{x}_j)\}_{j=1}^m = \{1 - 1/f'_k(\boldsymbol{x}_j)\}_{j=1}^m.$$

Then, we can obtain

$$\{f_k(\boldsymbol{x}_j)\}_{j=1}^m = \{f'_k(\boldsymbol{x}_j)\}_{j=1}^m,$$

which contradicts the condition that $\{f'_k(\boldsymbol{x}_j)\}_{j=1}^m$ is a different set. Therefore, the set of model outputs of the instance-level classifier $\{f_k(\boldsymbol{x}_j)\}_{j=1}^m$ is also determined uniquely, which concludes the proof. □

### B.4 Proof of Theorem 2

Before giving the proof, we introduce the following lemma.

**Lemma 3.** *Suppose that a proper loss function, e.g., the cross-entropy loss or the MSE loss is used, then we have $g_{k,l}^*(\mathbf{B}) = p(\tilde{\alpha}_k = l|\mathbf{B})$.*

*Proof.* **Cross-entropy loss.** Since the cross-entropy loss is always non-negative, minimizing $\tilde{R}_k(\boldsymbol{g}_k)$ can be achieved by minimizing the conditional risk $\mathbb{E}_{p(\tilde{\alpha}_k|\mathbf{B})}[\mathcal{L}(\boldsymbol{g}_k(\mathbf{B}), \tilde{\alpha}_k)]$ for every $\mathbf{B}$ (Masnadi-Shirazi & Vasconcelos, 2008). The conditional risk can be expressed as

$$-\sum_{l=1}^{m+1} p(\tilde{\alpha}_k = l|\mathbf{B}) \log(g_{k,l}(\mathbf{B})), \quad \text{s.t.} \sum_{l=1}^{m+1} g_{k,l}(\mathbf{B}) = 1.$$

We consider solving $\boldsymbol{g}_k^*(\mathbf{B})$ by using the Lagrange multiplier method (Bertsekas, 1997), which indicates minimizing

$$\phi = -\sum_{l=1}^{m+1} p(\tilde{\alpha}_k = l|\mathbf{B}) \log(g_{k,l}(\mathbf{B})) - \lambda \left( \sum_{l=1}^{m+1} g_{k,l}(\mathbf{B}) - 1 \right).$$

The partial derivative of $\phi$ w.r.t. $g_{k,l}(\mathbf{B})$ is

$$\frac{\partial \phi}{\partial g_{k,l}(\mathbf{B})} = -\frac{p(\tilde{\alpha}_k = l|\mathbf{B})}{g_{k,l}(\mathbf{B})} - \lambda.$$

By setting the partial derivative to zero, we have

$$g_{k,l}^*(\mathbf{B}) = -\frac{p(\tilde{\alpha}_k = l|\mathbf{B})}{\lambda}.$$

Besides, since

$$\sum_{l=1}^{m+1} g_{k,l}^*(\mathbf{B}) = -\frac{\sum_{l=1}^{m+1} p(\tilde{\alpha}_k = l|\mathbf{B})}{\lambda} = -\frac{1}{\lambda} = 1,$$

we have $\lambda = -1$. Therefore, we have $g_{k,l}^*(\mathbf{B}) = p(\tilde{\alpha}_k = l|\mathbf{B})$.

**MSE loss.** In a similar way, we minimize the conditional risk for every $\mathbf{B}$:

$$-\sum_{l=1}^{m+1} \left(g_{k,l}(\mathbf{B}) - p(\tilde{\alpha}_k = j|\mathbf{B})\right)^2, \quad \text{s.t.} \sum_{l=1}^{m+1} g_{k,l}(\mathbf{B}) = 1.$$

By using the Lagrange multiplier method, we aim at minimizing

$$\phi' = -\sum_{l=1}^{m+1} \left(g_{k,l}(\mathbf{B}) - p(\tilde{\alpha}_k = j|\mathbf{B})\right)^2 - \lambda \left(\sum_{l=1}^{m+1} g_{k,l}(\mathbf{B}) - 1\right).$$

The partial derivative of $\phi$ w.r.t. $g_{k,l}(\mathbf{B})$ is

$$\frac{\partial \phi'}{\partial g_{k,l}(\mathbf{B})} = -2g_{k,l}(\mathbf{B}) + 2p(\tilde{\alpha}_k = l|\mathbf{B}) - \lambda.$$

By setting the partial derivative to zero, we have

$$g_{k,l}^*(\mathbf{B}) = \frac{2p(\tilde{\alpha}_k = l|\mathbf{B}) - \lambda}{2}.$$

Besides, since

$$\sum_{l=1}^{m+1} g_{k,l}^*(\mathbf{B}) = \frac{2\sum_{l=1}^{m+1} p(\tilde{\alpha}_k = l|\mathbf{B}) - (m+1)\lambda}{2} = \frac{2 - (m+1)\lambda}{2} = 1,$$

we have $\lambda = 0$. Therefore, we have $g_{k,l}^*(\mathbf{B}) = p(\tilde{\alpha}_k = l|\mathbf{B})$. The proof is completed. $\qquad\square$

The we give the proof of Theorem 2.

*Proof of Theorem 2.* To begin with, we have

$$p(\tilde{\alpha}_k = l|\mathbf{B}) = \sum_{r=1}^{2^m} q_{r,l} \prod_{j=1}^m \left(p\left(\tilde{y}_j^k = 1|\boldsymbol{x}_j\right)^{b_j^{(r-1)}} p\left(\tilde{y}_j^k = 0|\boldsymbol{x}_j\right)^{\left(1-b_j^{(r-1)}\right)}\right),$$

$$g_{k,l}^*(\mathbf{B}) = \sum_{r=1}^{2^m} q_{r,l} \prod_{j=1}^m \left(f_k^*\left(\boldsymbol{x}_j\right)^{b_j^{(r-1)}} \left(1 - f_k^*\left(\boldsymbol{x}_j\right)\right)^{\left(1-b_j^{(r-1)}\right)}\right).$$

By Lemma 3 and Theorem 1, we have $\{f_k^*\left(\boldsymbol{x}_j\right)\}_{j=1}^m = \{p\left(\tilde{y}_j^k = 1|\boldsymbol{x}_j\right)\}_{j=1}^m$. When the labels are deterministic, we have $p(\tilde{y}_j^k = 1|\boldsymbol{x}_j) \in \{0, 1\}$. Therefore, we have $f_k^*\left(\boldsymbol{x}_j\right) \in \{0, 1\}$, which concludes the proof. $\qquad\square$

## B.5 PROOF OF THEOREM 3

First, we introduce an upper bound for loss $\mathcal{L}$. Assume that there exist constants $a$ and $b$ such that the model outputs of the binary classifier satisfy that $\forall k \in \{1, 2, \ldots, q\}, j \in \{1, 2, \ldots, m\}, 0 < a \leqslant f_k(\boldsymbol{x}_j) \leqslant b < 1$.

**Lemma 4.** *Assume that there exist constants $a$ and $b$ such that the model outputs of the binary classifier satisfy*

$$\forall k \in \{1, 2, \ldots, q\}, \; j \in \{1, 2, \ldots, m\}, \quad 0 < a \leqslant f_k(\boldsymbol{x}_j) \leqslant b < 1.$$

*When either the cross-entropy (CE) loss or the mean-squared-error (MSE) loss is used, we have*

$$\mathcal{L}\big(\boldsymbol{g}_k(\mathbf{B}), \tilde{\alpha}_k\big) = O(m). \tag{24}$$

*Proof.* We prove the statement separately for the two cases.

**Cross-entropy loss.**

$$\mathcal{L}_{\mathrm{CE}}\left(\boldsymbol{g}_k(\mathbf{B}), \tilde{\alpha}_k\right)$$

$$= -\log\left(g_{k,\tilde{\alpha}_k}(\mathbf{B})\right)$$

$$= -\log\left(\sum_{r=1}^{2^m} q_{r,\tilde{\alpha}_k} \prod_{j=1}^{m}\left(f_k\left(\boldsymbol{x}_j\right)^{b_j^{(r-1)}}\left(1-f_k\left(\boldsymbol{x}_j\right)\right)^{\left(1-b_j^{(r-1)}\right)}\right)\right)$$

$$\leqslant -\log\left(\sum_{r=1}^{2^m} q_{r,\tilde{\alpha}_k}\right)$$

$$-\frac{1}{\sum_{r=1}^{2^m} q_{r,\tilde{\alpha}_k}}\sum_{r=1}^{2^m} q_{r,\tilde{\alpha}_k}\log\left(\prod_{j=1}^{m}\left(f_k\left(\boldsymbol{x}_j\right)^{b_j^{(r-1)}}\left(1-f_k\left(\boldsymbol{x}_j\right)\right)^{\left(1-b_j^{(r-1)}\right)}\right)\right)$$

$$\leqslant -\frac{1}{\sum_{r=1}^{2^m} q_{r,\tilde{\alpha}_k}}\sum_{r=1}^{2^m} q_{r,\tilde{\alpha}_k}\sum_{j=1}^{m}\left(b_j^{(r-1)}\log\left(f_k\left(\boldsymbol{x}_j\right)\right)+\left(1-b_j^{(r-1)}\right)\log\left(1-f_k\left(\boldsymbol{x}_j\right)\right)\right)$$

$$\leqslant -\frac{1}{\sum_{r=1}^{2^m} q_{r,\tilde{\alpha}_k}}\sum_{r=1}^{2^m} q_{r,\tilde{\alpha}_k}\sum_{j=1}^{m}\left(b_j^{(r-1)}\log\left(a\right)+\left(1-b_j^{(r-1)}\right)\log\left(1-b\right)\right)$$

$$\leqslant -\frac{1}{\sum_{r=1}^{2^m} q_{r,\tilde{\alpha}_k}}\sum_{r=1}^{2^m} q_{r,\tilde{\alpha}_k}\, m\log(\min(a, 1-b))$$

$$= -m\log(\min(a, 1-b)) = O(m).$$

Here, the first inequality follows from Jensen's inequality.

**MSE Loss.**

$$\mathcal{L}_{\mathrm{MSE}}\left(\boldsymbol{g}_k(\mathbf{B}), \tilde{\alpha}_k\right) = \sum_{j=1}^{m+1}\left(\tilde{\alpha}_{k,j} - g_{k,j}(\mathbf{B})\right)^2 = \sum_{j=1}^{m+1}\left[\tilde{\alpha}_{k,j}(1-g_{k,j}(\mathbf{B}))^2 + (1-\tilde{\alpha}_{k,j})g_{k,j}(\mathbf{B})^2\right]$$

$$\leqslant \sum_{j=1}^{m+1}\max\left((1-a)^2, b^2\right) = (m+1)\max\left((1-a)^2, b^2\right) = O(m)$$

Combining both cases above, we conclude that the proof is completed. $\qquad\square$

**Lemma 5.** *Assume that there exist constants $a$ and $b$ such that the model outputs of the binary classifier satisfy*

$$\forall k \in \{1, 2, \ldots, q\},\ j \in \{1, 2, \ldots, m\}, \quad 0 < a \leqslant f_k(\boldsymbol{x}_j) \leqslant b < 1.$$

*For any two bag-level predictions $\boldsymbol{g}_k(\mathbf{B}), \boldsymbol{g}_k'(\mathbf{B})$ and the same one-hot target $\tilde{\alpha}_k$, the bound for the cross-entropy (CE) loss is*

$$\left|\mathcal{L}_{\mathrm{CE}}(\boldsymbol{g}_k(\mathbf{B}), \tilde{\alpha}_k) - \mathcal{L}_{\mathrm{CE}}(\boldsymbol{g}_k'(\mathbf{B}), \tilde{\alpha}_k)\right| \leqslant a^{-m}\left\|\boldsymbol{g}_k(\mathbf{B}) - \boldsymbol{g}_k'(\mathbf{B})\right\|_2,$$

*thus, the Lipschitz constant for CE is $a^{-m}$.*

*The bound for the mean-squared-error (MSE) loss is*

$$\left|\mathcal{L}_{\mathrm{MSE}}(\boldsymbol{g}_k(\mathbf{B}), \tilde{\alpha}_k) - \mathcal{L}_{\mathrm{MSE}}(\boldsymbol{g}_k'(\mathbf{B}), \tilde{\alpha}_k)\right| \leqslant 2\sqrt{2}\left\|\boldsymbol{g}_k(\mathbf{B}) - \boldsymbol{g}_k'(\mathbf{B})\right\|_2,$$

*thus, the Lipschitz constant for MSE is $2\sqrt{2}$.*

*Proof.* We prove the two loss functions separately.

**Cross-entropy loss.**

$$\mathcal{L}_{\mathrm{CE}}\left(\boldsymbol{g}_k(\mathbf{B}), \tilde{\alpha}_k\right) = -\log g_{k,\tilde{\alpha}_k}(\mathbf{B})$$

$$= -\log\left(\sum_{r=1}^{2^m} q_{r,\tilde{\alpha}_k}\prod_{j=1}^{m} f_k(\boldsymbol{x}_j)^{b_j^{(r-1)}}\left(1-f_k(\boldsymbol{x}_j)\right)^{1-b_j^{(r-1)}}\right).$$

Since each instance-level prediction satisfies $f_k(\boldsymbol{x}_j) \geqslant a$ and the weights $q_{r,\tilde{\alpha}_k}$ sum to one, it follows that

$$g_{k,\tilde{\alpha}_k}(\mathbf{B}) = \sum_r q_{r,\tilde{\alpha}_k}\prod_j f_k(\boldsymbol{x}_j)^{b_j^{(r-1)}}\left(1-f_k(\boldsymbol{x}_j)\right)^{1-b_j^{(r-1)}} \geqslant a^m.$$

Hence

$$\left|\nabla_{\boldsymbol{g}}\mathcal{L}_{\mathrm{CE}}\right| = \frac{1}{g_{k,\tilde{\alpha}_k}(\mathbf{B})} \leqslant a^{-m},$$

and for any two bag-level predictions $\boldsymbol{g}_k(\mathbf{B}), \boldsymbol{g}_k'(\mathbf{B})$,

$$\left|\mathcal{L}_{\mathrm{CE}}(\boldsymbol{g}_k(\mathbf{B}), \tilde{\alpha}_k) - \mathcal{L}_{\mathrm{CE}}(\boldsymbol{g}_k'(\mathbf{B}), \tilde{\alpha}_k)\right| \leqslant a^{-m} \left\|\boldsymbol{g}_k(\mathbf{B}) - \boldsymbol{g}_k'(\mathbf{B})\right\|_2.$$

Thus the Lipschitz constant for the CE loss is $a^{-m}$.

**MSE Loss.**

$$\mathcal{L}_{\mathrm{MSE}}\left(\boldsymbol{g}_k(\mathbf{B}), \tilde{\alpha}_k\right) = \sum_{\ell=0}^{m} \left(g_{k,\ell}(\mathbf{B}) - \tilde{\alpha}_{k,\ell}\right)^2 = \left\|\boldsymbol{g}_k(\mathbf{B}) - \tilde{\alpha}_k\right\|_2^2.$$

Its gradient is $\nabla_{\boldsymbol{g}}\mathcal{L}_{\mathrm{MSE}} = 2\left(\boldsymbol{g}_k(\mathbf{B}) - \tilde{\alpha}_k\right)$, so

$$\left\|\nabla_{\boldsymbol{g}}\mathcal{L}_{\mathrm{MSE}}\right\|_2 = 2\left\|\boldsymbol{g}_k(\mathbf{B}) - \tilde{\alpha}_k\right\|_2 \leqslant 2\sqrt{2},$$

because the maximum Euclidean distance between a probability vector and a one-hot vector in the simplex is $\sqrt{2}$. Therefore,

$$\left|\mathcal{L}_{\mathrm{MSE}}(\boldsymbol{g}_k(\mathbf{B}), \tilde{\alpha}_k) - \mathcal{L}_{\mathrm{MSE}}(\boldsymbol{g}_k'(\mathbf{B}), \tilde{\alpha}_k)\right| \leqslant 2\sqrt{2} \left\|\boldsymbol{g}_k(\mathbf{B}) - \boldsymbol{g}_k'(\mathbf{B})\right\|_2.$$

Hence the Lipschitz constant for the MSE loss is $2\sqrt{2}$. Both desired bounds are now established. $\square$

**Definition 2** (Rademacher complexity). Let $\tilde{\mathcal{D}}_k = \{(\mathbf{B}^i, \tilde{\alpha}_k^i)\}_{i=1}^n$ denote $n$ i.i.d. bags with proportional value labels drawn from a joint probability distribution $p(\mathbf{B}, \tilde{\alpha}_k)$. Let $\boldsymbol{\sigma} = (\sigma_1, \sigma_2, \ldots, \sigma_n)$ denote Rademacher variables taking values from $\{+1, -1\}$ uniformly. Besides, let $\mathcal{F}$ denote a class of binary classifiers. We introduce

$$\mathcal{G} = \left\{ \mathbf{B} \mapsto \boldsymbol{g}(\mathbf{B}) : \boldsymbol{g}(\mathbf{B}) = [g_1(\mathbf{B}), g_2(\mathbf{B}), \ldots, g_{m+1}(\mathbf{B})], g_l(\mathbf{B}) = \sum_{r=1}^{2^m} q_{r,l} \prod_{j=1}^{m} \right.$$
$$\left. \left( f\left(\boldsymbol{x}_j\right)^{b_j^{(r-1)}} \left(1 - f\left(\boldsymbol{x}_j\right)\right)^{\left(1 - b_j^{(r-1)}\right)} \right), f \in \mathcal{F} \right\}.$$

Notably, the marginal density of $\tilde{\mathcal{D}}_k, k \in \{1, 2, \ldots, q\}$ is the same. The (expected) Rademacher complexity of $\mathcal{L} \circ \mathcal{G}$ under the distribution $\tilde{\mathcal{D}}_k$ is defined as

$$\mathfrak{R}_n^k(\mathcal{L} \circ \mathcal{G}) = \mathbb{E}_{\tilde{\mathcal{D}}_k} \mathbb{E}_{\boldsymbol{\sigma}} \left[ \sup_{\boldsymbol{g} \in \mathcal{G}} \frac{1}{n} \sum_{i=1}^{n} \sigma_i \mathcal{L}\left(\boldsymbol{g}(\mathbf{B}^i), \tilde{\alpha}_k^i\right) \right].$$

For ease of notations, we introduce $\tilde{\mathcal{D}} = \tilde{\mathcal{D}}_1 \bigcup \tilde{\mathcal{D}}_2 \bigcup \ldots \bigcup \tilde{\mathcal{D}}_q$. Then, we introduce the following lemma.

**Lemma 6.** *The following inequality holds with probability at least $1 - \delta$, where $C$ is a constant:*

$$\sup_{f_1, f_2, \ldots, f_q \in \mathcal{F}} \left| \tilde{R}(f_1, f_2, \ldots, f_q) - \hat{\tilde{R}}(f_1, f_2, \ldots, f_q) \right| \leqslant \sum_{k=1}^{q} \mathfrak{R}_n^k(\mathcal{L} \circ \mathcal{G}) + Cm\sqrt{\frac{q \ln(2/\delta)}{2n}}.$$

*Proof.* In the following proofs, we consider a general case where all the datasets $\tilde{\mathcal{D}}_k$ are mutually independent. When a bag $(\mathbf{B}, \tilde{\alpha}_k)$ from $\tilde{\mathcal{D}}_k, k \in \{1, 2, \ldots, q\}$ is substituted by another bag $(\mathbf{B}', \tilde{\alpha}_k')$, according to Lemma 4, the value of $\sup_{f_1, f_2, \ldots, f_q \in \mathcal{F}} \left| \tilde{R}(f_1, f_2, \ldots, f_q) - \hat{\tilde{R}}(f_1, f_2, \ldots, f_q) \right|$ changes at most $Cm/n$. According to the McDiarmid's inequality, for any $\delta > 0$, the following inequality holds with probability at least $1 - \delta/2$:

$$\sup_{f_1, f_2, \ldots, f_q \in \mathcal{F}} \left( \tilde{R}(f_1, f_2, \ldots, f_q) - \hat{\tilde{R}}(f_1, f_2, \ldots, f_q) \right)$$
$$\leqslant \mathbb{E}_{\tilde{\mathcal{D}}} \left[ \sup_{f_1, f_2, \ldots, f_q \in \mathcal{F}} \left( \tilde{R}(f_1, f_2, \ldots, f_q) - \hat{\tilde{R}}(f_1, f_2, \ldots, f_q) \right) \right] + Cm\sqrt{\frac{q \ln(2/\delta)}{2n}}.$$

By symmetrization (Vapnik, 1998), it is routine work to have

$$\mathbb{E}_{\tilde{\mathcal{D}}}\left[\sup_{f_1,f_2,\ldots,f_q\in\mathcal{F}}\left(\tilde{R}(f_1,f_2,\ldots,f_q)-\hat{\tilde{R}}(f_1,f_2,\ldots,f_q)\right)\right]\leqslant 2\sum_{k=1}^q\mathfrak{R}_n^k(\mathcal{L}\circ\mathcal{G}).$$

Therefore, the following inequality holds with probability at least $1-\delta/2$:

$$\sup_{f_1,f_2,\ldots,f_q\in\mathcal{F}}\left(\tilde{R}(f_1,f_2,\ldots,f_q)-\hat{\tilde{R}}(f_1,f_2,\ldots,f_q)\right)\leqslant\sum_{k=1}^q\mathfrak{R}_n^k(\mathcal{L}\circ\mathcal{G})+Cm\sqrt{\frac{q\ln(2/\delta)}{2n}}.$$

In the same way, we have the following inequality with probability at least $1-\delta/2$:

$$\sup_{f_1,f_2,\ldots,f_q\in\mathcal{F}}\left(\hat{\tilde{R}}(f_1,f_2,\ldots,f_q)-\tilde{R}(f_1,f_2,\ldots,f_q)\right)\leqslant\sum_{k=1}^q\mathfrak{R}_n^k(\mathcal{L}\circ\mathcal{G})+Cm\sqrt{\frac{q\ln(2/\delta)}{2n}}.$$

Therefore, the following inequality holds with probability at least $1-\delta$:

$$\sup_{f_1,f_2,\ldots,f_q\in\mathcal{F}}\left|\tilde{R}(f_1,f_2,\ldots,f_q)-\hat{\tilde{R}}(f_1,f_2,\ldots,f_q)\right|\leqslant\sum_{k=1}^q\mathfrak{R}_n^k(\mathcal{L}\circ\mathcal{G})+Cm\sqrt{\frac{q\ln(2/\delta)}{2n}}.$$

$\square$

**Definition 3** (Covering number). Let $S=\{\boldsymbol{x}_1,\ldots,\boldsymbol{x}_m\}$ be a set of $m$ i.i.d. samples from $\mathcal{X}$ and define the empirical $L_p$ metric

$$\|f-g\|_{p,S}:=\left(\frac{1}{m}\sum_{\boldsymbol{x}\in S}|f(\boldsymbol{x})-g(\boldsymbol{x})|^p\right)^{1/p},\qquad p\geqslant 1.$$

An $\varepsilon$-*cover* of $\mathcal{F}$ (with respect to $\|\cdot\|_{p,S}$) is a finite set $\mathcal{C}\subseteq\mathcal{F}$ such that for every $f\in\mathcal{F}$ there exists $g\in\mathcal{C}$ with $\|f-g\|_{p,S}\leqslant\varepsilon$. The *empirical $L_p$ covering number* is the minimum size of such a cover,

$$\mathcal{N}_p(\mathcal{F},\varepsilon,S):=\min\{|\mathcal{C}|:\ \mathcal{C}\text{ is an }\varepsilon\text{-cover of }\mathcal{F}\text{ under }\|\cdot\|_{p,S}\},$$

and the *worst-case (over samples) covering number of size $m$* is

$$\mathcal{N}_p(\mathcal{F},\varepsilon,m):=\sup_{S\in\mathcal{X}^m}\mathcal{N}_p(\mathcal{F},\varepsilon,S).$$

**Lemma 7** (Refined Dudley's Inequality, Lemma A.1 from Srebro et al. (2010)). *For any function class $\mathcal{F}:\mathcal{X}\to\mathbb{R}$,*

$$\mathfrak{R}_n(\mathcal{F})\ \leqslant\ \inf_{\eta>0}\left\{4\eta\ +\ 12\int_\eta^{\sup_{f\in\mathcal{F}}\sqrt{\hat{\mathbb{E}}_n[f^2]}}\sqrt{\frac{\log\mathcal{N}_2(\mathcal{F},\varepsilon,n)}{n}}\,d\varepsilon\right\}.$$

**Lemma 8** (Theorem 18.4 in Anthony & Bartlett (2009)). *Let $\mathcal{F}$ be a nonempty set of real functions mapping from a domain $X$ into the real interval $[0,1]$ and suppose that $\mathcal{F}$ has finite pseudo-dimension $d$. Then for all $\varepsilon>0$,*

$$\mathcal{N}_1(\mathcal{F},\varepsilon,m)\ \leqslant\ \mathcal{M}_1(\mathcal{F},\varepsilon,m)\ <\ e\,(d+1)\left(\frac{2e}{\varepsilon}\right)^d.$$

**Corollary 1.** *Let $\mathcal{F}$ be a nonempty set of real functions mapping a domain $X$ into the interval $[0,1]$, and assume that $\mathcal{F}$ has finite pseudo-dimension $d$. Then, for every $\varepsilon>0$ and any sample size $m\geqslant 1$,*

$$\mathcal{N}_2(\mathcal{F},\varepsilon,m)\ \leqslant\ e\,(d+1)\left(\frac{2e}{\varepsilon^2}\right)^d.$$

*Proof.* Because all function values lie in $[0,1]$, for any $f,g\in\mathcal{F}$ we have the point-wise bound $|f(x)-g(x)|^2\leqslant|f(x)-g(x)|$. Averaging this inequality over an arbitrary set $S$ of $m$ i.i.d. instances and then taking square roots yields:

$$\|f-g\|_{2,S}^2\ \leqslant\ \|f-g\|_{1,S}\quad\Longrightarrow\quad\|f-g\|_{2,S}\ \leqslant\ \sqrt{\|f-g\|_{1,S}}. \tag{25}$$

If $\mathcal{C}$ is a $\delta$-cover of $\mathcal{F}$ in $L_1$, meaning that for every $f \in \mathcal{F}$ and every $S \in \mathcal{X}^m$, there exists a $g \in \mathcal{C}$ such that $\|f - g\|_{1,S} \leqslant \delta$, then by (25) the same center $g$ satisfies $\|f - g\|_{2,S} \leqslant \sqrt{\delta}$. Hence, any $\delta$-cover in $L_1$ induces a $\sqrt{\delta}$-cover in $L_2$:

$$\mathcal{N}_2\big(\mathcal{F}, \sqrt{\delta}, m\big) \leqslant \mathcal{N}_1\big(\mathcal{F}, \delta, m\big). \tag{26}$$

Choose $\delta = \varepsilon^2$ in (26) and invoke Lemma 8, we have

$$\mathcal{N}_2\big(\mathcal{F}, \varepsilon, m\big) \leqslant \mathcal{N}_1\big(\mathcal{F}, \varepsilon^2, m\big) < e\,(d+1)\Big(\tfrac{2e}{\varepsilon^2}\Big)^d,$$

which is exactly the claimed inequality. $\qquad\square$

**Lemma 9.** *Let $\mathcal{F} \subseteq [0,1]^{\mathcal{X}}$ be a class of real-valued functions with pseudo-dimension $d$, and let $\mathcal{L}$ be any loss function whose range is contained within $[0,T]$, and with Lipschitz constant $\lambda$ with respect to $\boldsymbol{g}$. Denote by $\mathcal{L} \circ \mathcal{G}$ the composition of $\mathcal{L}$ and the class $\mathcal{G}$ as defined in Definition 2. Then, for any sample of $n$ bags under the distribution $\tilde{\mathcal{D}}_k$,*

$$\mathfrak{R}_n^k(\mathcal{L} \circ \mathcal{G}) = \tilde{O}\big(T\sqrt{\tfrac{d(m + \log \lambda)}{n}}\big),$$

*where $\tilde{O}(\cdot)$ hides universal constants and all polynomial dependencies on $\log m$ and $\log n$.*

*Proof.* For any pair of functions $f$ and $f^\epsilon$, and their corresponding bag-level mappings $\boldsymbol{g}$ and $\boldsymbol{g}^\epsilon$, we first note that for every bag $\mathbf{B}^i$ and each $l$, according to the definition of $\boldsymbol{g}$, we have:

$$|g_l(\mathbf{B}^i) - g_l^\epsilon(\mathbf{B}^i)|$$

$$\leqslant \left| \sum_{\substack{S \subseteq [m] \\ |S|=l}} \Big(\prod_{j \in S} f(\boldsymbol{x}_{i,j})\Big)\Big(\prod_{j \notin S}(1 - f(\boldsymbol{x}_{i,j}))\Big) - \sum_{\substack{S \subseteq [m] \\ |S|=l}} \Big(\prod_{j \in S} f^\epsilon(\boldsymbol{x}_{i,j})\Big)\Big(\prod_{j \notin S}(1 - f^\epsilon(\boldsymbol{x}_{i,j}))\Big) \right|$$

$$\leqslant \sum_{\substack{S \subseteq [m] \\ |S|=l}} \sum_{j=1}^m \big|f(\boldsymbol{x}_{i,j}) - f^\epsilon(\boldsymbol{x}_{i,j})\big|$$

$$= \binom{m}{\ell} \sum_{j=1}^m \big|f(\boldsymbol{x}_{i,j}) - f^\epsilon(\boldsymbol{x}_{i,j})\big|$$

$$\leqslant 2^m \sum_{j=1}^m \big|f(\boldsymbol{x}_{i,j}) - f^\epsilon(\boldsymbol{x}_{i,j})\big|$$

$$\leqslant 2^m \sqrt{m} \sqrt{\sum_{j=1}^m \big(f(\boldsymbol{x}_{i,j}) - f^\epsilon(\boldsymbol{x}_{i,j})\big)^2}$$

$$= 2^m m \sqrt{\tfrac{1}{m} \sum_{j=1}^m \big(f(\boldsymbol{x}_{i,j}) - f^\epsilon(\boldsymbol{x}_{i,j})\big)^2}.$$

The first inequality applies the triangle inequality. The second inequality uses the bound on individual differences due to each term being at most 1, leveraging the boundedness on $f(\boldsymbol{x}_{i,j})$ and $f^\epsilon(\boldsymbol{x}_{i,j})$. The fourth inequality employs the combinatorial bound $\binom{m}{\ell} \leqslant 2^m$. Finally, the fifth inequality applies the Cauchy–Schwarz inequality to convert from the $\ell_1$-norm to the $\ell_2$-norm.

Consequently, taking the $\ell_2$-norm over the entire $(m+1)$-dimensional output vector and then applying the standard relation $\|v\|_2 \leqslant \sqrt{m+1}\|v\|_\infty$, we obtain

$$\|\boldsymbol{g}(\mathbf{B}^i) - \boldsymbol{g}^\epsilon(\mathbf{B}^i)\|_2 \leqslant \sqrt{m+1}\,\|\boldsymbol{g}(\mathbf{B}^i) - \boldsymbol{g}^\epsilon(\mathbf{B}^i)\|_\infty \leqslant 2^{2m}\sqrt{\tfrac{1}{m}\sum_{j=1}^m \big(f(\boldsymbol{x}_{i,j}) - f^\epsilon(\boldsymbol{x}_{i,j})\big)^2},$$

in the last inequality we deliberately absorbed the polynomial factor into the exponential term. Hence, for the empirical $\ell_2$-metrics over the sample we have the compact bound

$$\left(\frac{1}{n}\sum_{i=1}^{n}\left|\mathcal{L}\big(\boldsymbol{g}(\mathbf{B}^i),\tilde{\alpha}_k^i\big)-\mathcal{L}\big(\boldsymbol{g}^\epsilon(\mathbf{B}^i),\tilde{\alpha}_k^i\big)\right|^2\right)^{1/2}\leqslant\lambda\left(\frac{1}{n}\sum_{i=1}^{n}\|\boldsymbol{g}(\mathbf{B}^i)-\boldsymbol{g}^\epsilon(\mathbf{B}^i)\|_2^2\right)^{1/2}\leqslant$$

$$\lambda\,2^{2m}\left(\frac{1}{nm}\sum_{i=1}^{n}\sum_{j=1}^{m}\big(f(\boldsymbol{x}_{i,j})-f^\epsilon(\boldsymbol{x}_{i,j})\big)^2\right)^{1/2}.$$

Then, we can directly derive the following relationship between covering numbers:

$$\log\mathcal{N}_2\big(\mathcal{L}\circ\mathcal{G},\varepsilon,n\big)\leqslant\log\mathcal{N}_2\big(\mathcal{G},\tfrac{\varepsilon}{\lambda},n\big)\leqslant\log\mathcal{N}_2\big(\mathcal{F},\tfrac{\varepsilon}{\lambda\,2^{2m}},nm\big).$$

Applying Corollary 1 to the right–hand covering number:

$$\log\mathcal{N}_2\big(\mathcal{L}\circ\mathcal{G},\varepsilon,n\big)\leqslant\log\big(e(d+1)\big)+d\,\log\Big(\tfrac{2e\,\lambda\,2^{2m}}{\varepsilon^2}\Big).$$

Because the outputs of $\mathcal{L}$ lie in $[0,T]$, we can apply Lemma 7, which yields:

$$\mathfrak{R}_n^k(\mathcal{L}\circ\mathcal{G})\leqslant\inf_{\eta>0}\left\{4\eta+12\int_\eta^T\sqrt{\frac{\log\big(e(d+1)\big)+2d\,\log\Big(\frac{\sqrt{2e}\lambda\,2^m}{\varepsilon}\Big)}{n}}\,d\varepsilon\right\}.$$

Choosing $\eta=\Theta\big(T\sqrt{2d/n}\big)$ gives

$$\mathfrak{R}_n^k(\mathcal{L}\circ\mathcal{G})\leqslant4T\sqrt{\frac{2d}{n}}+12T\sqrt{\frac{6dm+d\log\lambda+d\log n}{n}}.$$

We absorb the $\log n$ and $\log\lambda$ factors into the $\tilde{O}$ notation and, using $m\geqslant1$, obtain

$$\mathfrak{R}_n^k(\mathcal{L}\circ\mathcal{G})=\tilde{O}\big(T\sqrt{\frac{d(m+\log\lambda)}{n}}\big),$$

which concludes the proof. $\qquad\square$

**Corollary 2.** *Suppose we use either the cross-entropy (CE) loss or the mean-squared-error (MSE) loss. Then, under the assumptions stated in Lemma 9, we have for any sample of $n$ bags under the distribution $\tilde{\mathcal{D}}_k$,*

$$\mathfrak{R}_n^k(\mathcal{L}\circ\mathcal{G})=\tilde{O}\left(\sqrt{\frac{dm^3}{n}}\right),$$

*where $\tilde{O}(\cdot)$ hides universal constants and polynomial dependencies on $\log m$ and $\log n$.*

*Proof.* The corollary can be obtained by combining Lemmas 4 and 5. $\qquad\square$

Finally, we provide the proof of Theorem 3.

*Proof of Theorem 3.*
$$\tilde{R}(\hat{f}_1,\hat{f}_2,\dots,\hat{f}_q)-\tilde{R}(f_1^*,f_2^*,\dots,f_q^*)$$
$$=\tilde{R}(\hat{f}_1,\hat{f}_2,\dots,\hat{f}_q)-\hat{\tilde{R}}(\hat{f}_1,\hat{f}_2,\dots,\hat{f}_q)+\hat{\tilde{R}}(\hat{f}_1,\hat{f}_2,\dots,\hat{f}_q)-\hat{\tilde{R}}(f_1^*,f_2^*,\dots,f_q^*)$$
$$+\hat{\tilde{R}}(f_1^*,f_2^*,\dots,f_q^*)-\tilde{R}(f_1^*,f_2^*,\dots,f_q^*)$$
$$\leqslant\tilde{R}(\hat{f}_1,\hat{f}_2,\dots,\hat{f}_q)-\hat{\tilde{R}}(\hat{f}_1,\hat{f}_2,\dots,\hat{f}_q)+\hat{\tilde{R}}(f_1^*,f_2^*,\dots,f_q^*)-\tilde{R}(f_1^*,f_2^*,\dots,f_q^*)$$
$$\leqslant2\sup_{f_1,f_2,\dots,f_q\in\mathcal{F}}\left|\tilde{R}(f_1,f_2,\dots,f_q)-\hat{\tilde{R}}(f_1,f_2,\dots,f_q)\right|\qquad(27)$$

The first inequality is deduced because $(\hat{f}_1,\hat{f}_2,\dots,\hat{f}_q)$ is the minimizer of $\hat{\tilde{R}}(f_1,f_2,\dots,f_q)$. By combining Inequality (27) with Lemma 6 and Corollary 2, the proof is completed. $\qquad\square$

### B.6  PROOF OF PROPOSITION 1

We denote $T^{\text{arith}}$ as the arithmetic-operation complexity and $T^{\text{wall}}$ as the wall-clock-time complexity.

**(i) Initial forward transforms (GPU batch).** All $m + 1$ zero-padded binomials undergo one *batched* discrete Fourier transform call. Since each discrete Fourier transform requires $O(m \log m)$ time (Cooley & Tukey, 1965), we have:

$$T_{\text{init}}^{\text{arith}} = (m + 1)\, O\big((m + 1) \log(m + 1)\big) = O(m^2 \log m), \qquad T_{\text{init}}^{\text{wall}} = O(m \log m),$$

because a single batched call executes all vectors in parallel on GPU, leveraging the high-throughput parallelism provided by modern GPU-accelerated FFT libraries.

**(ii) Layer–$i$ merges in the frequency domain.** After $i$ recursions the list length is $N_i = \lceil (m + 1)/2^i \rceil$. For each even $j < N_i - 1$ we set

$$H_{k,j,t}^{s} = H_{k,2j-1,t}^{s-1}\, H_{k,2j,t}^{s-1}, \quad t = 0, \ldots, m,$$

which costs $m + 1 = O(m)$ arithmetic operations per pair. Launching the $\lfloor N_i/2 \rfloor$ pairs of layer $i$ as a *single* GPU kernel yields the wall-clock cost

$$T_i^{\text{wall}} = O(m).$$

Therefore

$$T_{\text{merge}}^{\text{wall}} = \sum_{i=0}^{\lceil \log_2(m+1) \rceil - 1} T_i^{\text{wall}} = O(m \log m),$$

whereas the arithmetic count is

$$T_{\text{merge}}^{\text{arith}} = \sum_{i=0}^{\log_2 m} \left\lfloor \frac{N_i}{2} \right\rfloor O(m) = O(m^2).$$

**(iii) Final inverse transform (GPU).** A single length–$(m + 1)$ inverse discrete Fourier transform (IDFT) yields $\{c_{k,0}, \ldots, c_{k,m}\}$ and hence $g_{k,l} = c_{k,l-1}$. Since the inverse transform has the same complexity as the forward transform (Cooley & Tukey, 1965), we have:

$$T_{\text{final}}^{\text{arith}} = T_{\text{final}}^{\text{wall}} = O(m \log m).$$

**(iv) Overall wall-clock time.**

$$T_{\text{total}}^{\text{wall}} = T_{\text{init}}^{\text{wall}} + T_{\text{merge}}^{\text{wall}} + T_{\text{final}}^{\text{wall}} = O(m \log m) + O(m \log m) + O(m \log m) = O\big(m \log m\big).$$

With fixed length–$(m + 1)$ padding and batched GPU kernels that keep all merges in the frequency domain, the procedure runs in $O(m \log m)$ wall-clock time, and returns the desired coefficients $c_{k,r}$ and probabilities $g_{k,l}(\mathbf{B}) = c_{k,l-1}$. $\qquad\square$

## C  MORE ALGORITHMIC DETAILS

The algorithmic details of LLP-PVC, the FFT-based computation of model outputs, the cluster bag generation process, and the proportion-first bag generation process are shown in Algorithms 1 through 4. The pipeline of LLP-PVC and a toy example of FFT-based calculation of posterior probabilities for proportion values is shown in Figures 3 and 4, respectively.

---

**Algorithm 2** FFT-based computation of $\boldsymbol{g}_k(\mathbf{B})$ (Algorithm 2)

---

**Require:** Bag $\mathbf{B} = \{\boldsymbol{x}_1, \ldots, \boldsymbol{x}_m\}$; per-class instance posteriors $\{f_k(\boldsymbol{x}_j)\}_{j=1}^m$; $N \leftarrow m + 1$.
**Ensure:** $\boldsymbol{g}_k(\mathbf{B}) = [g_{k,1}(\mathbf{B}), \ldots, g_{k,N}(\mathbf{B})]$ with $g_{k,l}(\mathbf{B}) = c_{k,l-1}$.
 1: **for** $j = 1$ **to** $m$ **do**
 2:     Initialize $\boldsymbol{h}_{k,j}^{(0)} \leftarrow [\, 1 - f_k(\boldsymbol{x}_j),\ f_k(\boldsymbol{x}_j),\ 0, \ldots, 0\,] \in \mathbb{R}^N$   (according to Eq. (15))
 3:     $\boldsymbol{H}_{k,j}^{(0)} \leftarrow \mathrm{DFT}(\boldsymbol{h}_{k,j}^{(0)})$   using Eq. (18)
 4: **end for**
 5: $s \leftarrow 1$, $N_1 \leftarrow m$
 6: **while** $N_s > 1$ **do**
 7:     **for** $j = 1$ **to** $\lfloor N_s/2 \rfloor$ **do**
 8:         $\boldsymbol{H}_{k,j}^{s} \leftarrow \boldsymbol{H}_{k,2j-1}^{s-1} \odot \boldsymbol{H}_{k,2j}^{s-1}$   using Eq. (19)
 9:     **end for**
10:     **if** $N_s$ is odd **then**
11:         $\boldsymbol{H}_{k, \frac{N_s+1}{2}}^{s} \leftarrow \boldsymbol{H}_{k,N_s}^{s-1}$   as in Eq. (19)
12:     **end if**
13:     $s \leftarrow s + 1$; $N_s \leftarrow \lceil N_{s-1}/2 \rceil$
14: **end while**
15: $\boldsymbol{H}_k^* \leftarrow \boldsymbol{H}_{k,1}^{\lceil \log_2(m+1) \rceil}$   according to Eq. (16)
16: $\boldsymbol{c}_k \leftarrow \mathrm{IDFT}(\boldsymbol{H}_k^*)$   using Eq. (20)
17: **for** $l = 1$ **to** $N$ **do**
18:     $g_{k,l}(\mathbf{B}) \leftarrow c_{k,l-1}$   by Lemma 2
19: **end for**
20: **return** $\boldsymbol{g}_k(\mathbf{B})$

---

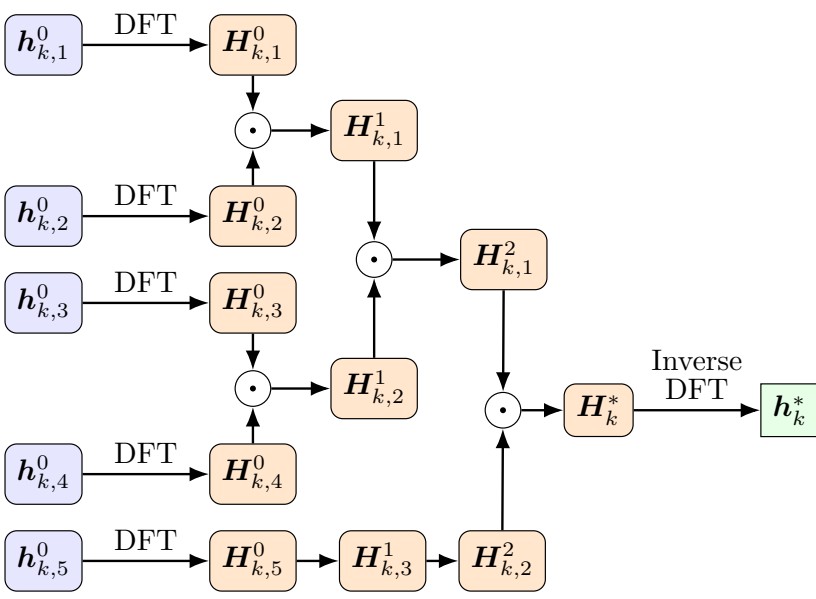

Figure 4: A toy example of FFT-based calculation of posterior probabilities for proportion values.

---

**Algorithm 3** Cluster bag generation

---

**Require:** Labeled dataset $\{(\boldsymbol{x}_i, y_i)\}_{i=1}^N$ with $y_i \in \mathcal{Y} = \{1, \ldots, q\}$; bag size $m$; number of clusters $K$; Dirichlet parameter $\boldsymbol{\pi} \in \mathbb{R}_+^K$.
**Ensure:** LLP dataset $\mathcal{D} = \{(\mathbf{B}_i, \boldsymbol{\alpha}_i)\}_{i=1}^n$, where $n = \lfloor N/m \rfloor$.
1: $n \leftarrow \lfloor N/m \rfloor$; $N' \leftarrow nm$; keep the first $N'$ samples.
2: Partition $\{\boldsymbol{x}_i\}_{i=1}^{N'}$ into $K$ clusters; let $\mathrm{clu}[r] \leftarrow \{i \leqslant N' : \mathrm{cluster}(i) = r\}$ for $r = 1, \ldots, K$.
3: Initialize $\mathcal{D} \leftarrow \varnothing$.
4: **for** $i = 1$ **to** $n$ **do**
5:     Draw cluster proportions $\boldsymbol{\rho}_i \sim \mathrm{Dirichlet}(\boldsymbol{\pi})$.                 $\triangleright \boldsymbol{\rho}_i \in \Delta^{K-1}$
6: **end for**
7: **for** $i = 1$ **to** $n$ **do**
8:     Obtain an index list $I^i$ by drawing $m$ indices from $\bigcup_{r=1}^K \mathrm{clu}[r]$ *without replacement*, such that the cluster composition of $I^i$ follows $\boldsymbol{\rho}_i$ and the global usage exhausts each $\mathrm{clu}[r]$. $\triangleright I^i$ is the list of selected sample indices for bag $i$
9:     Form the bag $\mathbf{B}_i = [\boldsymbol{x}_{i1}, \ldots, \boldsymbol{x}_{im}]$ from indices $I^i$.
10:     Compute label proportions $\boldsymbol{\alpha}_i = [\alpha_{i1}, \ldots, \alpha_{iq}]$ where $\alpha_{ik} \leftarrow \frac{1}{m} \sum_{j=1}^m \mathbb{I}(y_{ij} = k)$ for $k = 1, \ldots, q$.
11:     Add $(\mathbf{B}_i, \boldsymbol{\alpha}_i)$ to $\mathcal{D}$.
12: **end for**
13: **return** $\mathcal{D}$

---

**Algorithm 4** $\alpha$-First bag generation

---

**Require:** Labeled dataset $\{(\boldsymbol{x}_i, y_i)\}_{i=1}^N$ with $y_i \in \mathcal{Y} = \{1, \ldots, q\}$; bag size $m$; Dirichlet parameter $\boldsymbol{\pi} \in \mathbb{R}_+^K$.
**Ensure:** LLP dataset $\mathcal{D} = \{(\mathbf{B}_i, \boldsymbol{\alpha}_i)\}_{i=1}^n$, where $n = \lfloor N/m \rfloor$.
1: $n \leftarrow \lfloor N/m \rfloor$; $N' \leftarrow nm$; keep the first $N'$ samples.
2: Build class pools $\mathrm{pool}[k] \leftarrow \{i \leqslant N' : y_i = k\}$ for $k = 1, \ldots, q$.
3: Initialize $\mathcal{D} \leftarrow \varnothing$.
4: **for** $i = 1$ **to** $n$ **do**
5:     Draw target proportions $\boldsymbol{\alpha}_i \sim \mathrm{Dirichlet}(\boldsymbol{\pi})$.
6: **end for**
7: **for** $i = 1$ **to** $n$ **do**
8:     Obtain an index list $I^i$ by drawing $m$ indices from $\bigcup_{k=1}^q \mathrm{pool}[k]$ *without replacement*, such that the class composition of $I^i$ follows $\boldsymbol{\alpha}_i$ and the global usage exhausts each pool.     $\triangleright I^i$ is the list of selected sample indices for bag $i$
9:     Form $\mathbf{B}_i = [\boldsymbol{x}_{i1}, \ldots, \boldsymbol{x}_{im}]$ from indices $I^i$.
10:     Add $(\mathbf{B}_i, \boldsymbol{\alpha}_i)$ to $\mathcal{D}$.
11: **end for**
12: **return** $\mathcal{D}$

---

# D   MORE EXPERIMENTAL DETAILS AND RESULTS

## D.1   MORE EXPERIMENTAL DETAILS

The hyperparameters used for K-MNIST, F-MNIST, SVHN, and CIFAR-10 are described in Table 6. Furthermore, for numerical stability, we clip the probabilities in Eq. (14) by enforcing a minimum value of $\varepsilon$ during optimization, where $\varepsilon \in \{10^{-12}, 10^{-30}, 10^{-80}, 10^{-200}\}$.

Table 6: Hyperparameters for datasets (K-MNIST, F-MNIST, SVHN, CIFAR-10). SVHN and CIFAR-10 backbones ResNet-18 are pretrained on ImageNet.

| Dataset | K-MNIST | F-MNIST | SVHN | CIFAR-10 |
|---|---|---|---|---|
| Model | LeNet-5 | | ResNet-18 | |
| Epoch | 1024 | | 500 | |
| Learning Rate | $2.5 \times 10^{-3}$ | | $5 \times 10^{-4}$ | |
| Weight Decay | $5 \times 10^{-5}$ | | $1 \times 10^{-4}$ | |
| Optimizer | SGD (momentum 0.9) | | | |
| Data Augmentation | Random crop & horizontal flip | | | |
| Batch Size | 1024 | | | |
| Scheduler | $\eta = \eta_0 \cos\left(\frac{7\pi k}{16K}\right)$ | | | |

## D.2 ADDITIONAL EXPERIMENTS WITH LARGE BAG SIZES

We also evaluate our method under large bag sizes, and experimental results are shown in Table 7. While prior LLP studies typically consider even larger bags, mainly in binary settings (e.g., up to $m=2048$), extending to multi-class classification is substantially more challenging. In our experiments, meaningful performance is achieved at $m = 256$, whereas for larger $m$, most baselines quickly collapse toward trivial solutions, rendering further increases in bag size less informative.

Table 7: Accuracy results (mean $\pm$ std over seeds) under **Random Bag** construction with **large bag sizes**.

| Method | CIFAR-10 | | | SVHN | | |
|---|---|---|---|---|---|---|
| | 256 | 512 | 1024 | 256 | 512 | 1024 |
| PM | $35.17 \pm 4.30$ | $21.20 \pm 4.46$ | $9.82 \pm 2.52$ | $28.14 \pm 1.40$ | $20.65 \pm 1.62$ | $15.64 \pm 1.52$ |
| DSQ | $42.35 \pm 2.45$ | $\mathbf{32.83 \pm 2.96}$ | $10.39 \pm 3.12$ | $35.16 \pm 0.63$ | $\mathbf{25.97 \pm 0.08}$ | $15.63 \pm 2.11$ |
| EasyLLP | $31.11 \pm 1.43$ | $22.95 \pm 1.54$ | $10.77 \pm 0.93$ | $24.69 \pm 1.45$ | $21.82 \pm 0.94$ | $19.59 \pm 0.00$ |
| EasyLLP-flood | $39.07 \pm 1.62$ | $28.22 \pm 0.91$ | $10.76 \pm 0.93$ | $36.02 \pm 2.07$ | $25.20 \pm 1.05$ | $19.59 \pm 0.00$ |
| ROT | $37.17 \pm 1.56$ | $22.60 \pm 2.98$ | $8.39 \pm 0.11$ | $32.79 \pm 0.52$ | $21.40 \pm 1.99$ | $15.04 \pm 1.16$ |
| LLP-PVC | $\mathbf{69.21 \pm 1.65}$ | $28.46 \pm 5.82$ | $\mathbf{12.92 \pm 1.98}$ | $\mathbf{90.74 \pm 2.32}$ | $21.03 \pm 1.23$ | $\mathbf{19.62 \pm 0.04}$ |

