# OpenReview forum: "Learning from Label Proportions via Proportional Value Classification"
_ICLR.cc/2026/Conference — ICLR 2026 Poster_

### Official Review · Reviewer_n3aR · 2025-10-22

**Soundness:** 3
**Presentation:** 3
**Contribution:** 3
**Rating:** 6
**Confidence:** 4

**Summary:**

This paper proposes a novel approach for LLP, a weakly supervised learning paradigm where the training data consists of bags of instances annotated only with the overall proportion of each label within the bag, and the goal is to learn an instance-level classifier. The authors identify a limitation in prevailing proportion matching strategies: over-smoothing problem, where the model outputs for all instances in a bag converge indistinguishably towards the proportion value, resulting in poor instance-level classification performance. To address this, the paper introduces PVC, a method that reframes the LLP problem. Instead of fitting the proportions directly, the label proportions are treated as targets for an auxiliary multi-class classification task. This auxiliary task involves predicting the specific proportional value from the bag of instances.

**Strengths:**

- This paper establishes a theoretical bridge between the posterior probabilities of the bag-level proportional values and the instance-level labels, enabling the induction of the target instance-level classifier.
The computational algorithm based on a divide-and-conquer strategy is efficient, achieving a time complexity of $O(m \log m)$, which is significantly more efficient than a dynamic programming baseline with $O(m^2)$ complexity.
- The authors prove that under specific conditions, particularly when labels are deterministic, the outputs of the optimal instance-level classifier will converge to 0 or 1. The authors claim this property reveals that the proposed PVC mitigates the over-smoothing problem. I have checked the overall logical soundness of the proofs but have not delved into a detailed verification of the specific derivations.

**Weaknesses:**

1. The core claim of this paper is that PVC mitigates the over-smoothing problem. The authors present Theorems 1 and 2 as the theoretical foundation for this claim. While Theorem 1 establishes a one-to-one correspondence between the bag-level and instance-level classifier outputs, and Theorem 2 states that the optimal instance-level classifier outputs will be 0 or 1 under deterministic labels, the logical connection from these theorems to the concrete mitigation of over-smoothing remains inadequately articulated. It is unclear how these theoretical results, in a step-by-step manner, directly lead to the conclusion that PVC prevents the over-smooth.

2. The output layers of neural network classifiers typically employ sigmoid or softmax functions to obtain label probabilities. Directly increasing the temperature coefficient in these functions can also make the model more inclined to output extreme values of 0 or 1. This straightforward strategy appears to offer an alternative approach to addressing the over-smoothing problem. So, compared to the temperature-based method, the merits of PVC is not clear.

3. The experimental validation is insufficient to robustly support the authors' central claim. Although the method demonstrates competitive accuracy, the claim of effectively alleviating over-smoothing is not thoroughly verified. The evidence for alleviation of over-smooth is primarily limited to the four figures presented in the introduction. Besides, relying solely on accuracy as the primary evaluation metric is also insufficient for a classification problem.

**Questions:**

1. Could you please provide a more detailed explanation of the physical insights behind Theorems 1 and 2, clarifying how they theoretically demonstrate that PVC mitigates the over-smoothing problem?

2. Could you please elaborate on the comparative advantages of PVC over direct temperature scaling, and include an experimental analysis of whether their performance difference is significant?

3. Could you please supply additional experimental evidence demonstrating how PVC alleviates over-smoothing, and present results on metrics such as precision and recall?

---

> ### Author Response · Authors · 2025-11-22
>
> We sincerely appreciate your time and effort in reviewing our paper.
>
> **Response to Q1: Further Explanation of Why Our Method Mitigates Over-Smoothing**
>
> The problems caused by oversmoothing due to proportion matching are due to the trivial solution that the model's outputs for individual instances naively fit the proportion value of the bag. This results in severe performance degradation. To solve this problem, Theorem 1 of our paper establishes a one-to-one correspondence between bag-level and instance-level classifier outputs by introducing the PVC task. If the cross-entropy or mean squared error (MSE) loss is then used, the optimal instance-level classifier outputs will be the posterior probabilities and will be 0 or 1 in deterministic cases. This differs from the given proportion value of the bag. The sharpened classifier outputs demonstrate the classifier's confidence in whether an instance belongs to a class or not, leading to smaller entropy values and better classification performance.
>
> **Response to Q2: Whether Temperature Scaling Can Directly Address Overfitting**
>
> Thank you for the suggestion regarding the use of a temperature parameter. Compared with our method, we see two main differences. First, as shown for example in Theorem 2, in the deterministic case where $p(\tilde{y}\_j^{k} = 1\mid\mathbf{x}\_{j}) \in \\{0,1\\}$
> , our learned classifier satisfies $f^{\star}\_{k}(\mathbf{x}\_{j}) \in \\{0,1\\}$.
> In contrast, in the non-deterministic case, our model does not become increasingly sharp. Second, our PVC method leverages the proportional-value supervision: under the proportion constraints, optimizing Eq. (10) gradually drives the model away from overly smooth solutions. A simple temperature adjustment cannot achieve these advantages—it only increases the output sharpness uniformly without using any proportional supervision. Moreover, introducing a temperature parameter adds an extra hyperparameter that may require additional tuning across different datasets and different bag sizes.
>
> Below are the results obtained on AgNews with bag size = 128 under the PM method, where we apply several commonly used temperature values to sharpen the model’s output distribution. As shown, lowering the temperature does not effectively alleviate the problem; instead, it leads to degraded performance in accuracy, precision, and recall. We suspect that temperature scaling alters the output distribution in a way that disrupts the model’s convergence, ultimately harming performance. From the results, we can see that using a single temperature does not resolve the over-smoothing of the model outputs.
>
> | temperature | entropy | accuracy | precision | recall |
> |-|-|-|-|-|
> | 0.2 | 1.3861 | 46.5205 | 0.4648 | 0.4653 |
> | 0.5 | 1.3862 | 59.2568 | 0.6248 | 0.5926 |
> | 0.8 | 1.3862 | 65.9751 | 0.6713 | 0.6545 |
> | 1 | 1.3861 | 67.4310 | 0.6823 | 0.6739 |
>
> **Response to Q3: Further Experiments on Alleviating Over-Smoothing and Expanding Evaluation Metrics**
>
> We conduct experiments on the AGNews and Yelp datasets, reporting accuracy, precision, recall, and prediction entropy. These results demonstrate that our method not only substantially improves classification performance, but also mitigates over-smoothing: compared with existing baselines, LLP-PVC achieves higher accuracy, precision, and recall while simultaneously producing significantly lower entropy. We will include these results in the final paper.
>
> Agnews:
>
> | Method | entropy | accuracy | precision | recall |
> |-|-|-|-|-|
> | PM | 1.3862 | 67.34 | 0.6853 | 0.6701 |
> | DSQ | 1.3841 | 67.62 | 0.6861 | 0.6714 |
> | EasyLLP | 1.3857 | 67.27 | 0.6785 | 0.6729 |
> | EasyLLP_flood | 1.3862 | 67.27 | 0.6785 | 0.6729 |
> | ROT | 1.3855 | 67.48 | 0.6804 | 0.6750 |
> | LLP-PVC | 0.0726 | **88.84** | **0.8887** | **0.8884** |
>
> Yelp:
>
> | Method | entropy | accuracy | precision | recall |
> |-|-|-|-|-|
> | PM | 1.6091 | 45.41 | 0.4354 | 0.4392 |
> | DSQ | 1.6041 | 45.00 | 0.4296 | 0.4323 |
> | EasyLLP | 1.6091 | 45.19 | 0.4353 | 0.4389 |
> | EasyLLP_flood | 1.6084 | 45.22 | 0.4352 | 0.4401 |
> | ROT | 1.6090 | 45.30 | 0.4265 | 0.4300 |
> | LLP-PVC | 0.2381 | **56.87** | **0.5660** | **0.5688** |

---

### Official Review · Reviewer_B5vC · 2025-10-26

**Soundness:** 3
**Presentation:** 2
**Contribution:** 2
**Rating:** 4
**Confidence:** 4

**Summary:**

The paper introduces a method to solve the learning from the label proportions (LLP) problem, where we'd like to obtain an instance-level classifier given aggregated bag-level data. The proposed method claims to mitigate the oversmoothing problem in the proportion matching strategy. This work introduces a new algorithm for LLP accompanied by theoretical analysis and empirical validations.

**Strengths:**

1. The method reduces LLP to a bag-level classification problem, while prior work on proportion matching typically maps it to a bag-level classification problem.

2. Theorem 2 points out that the optimal solution of the proposed bag-level objective leads to an optimal instance-level classifier. Theorem 3 shows that the optimizer of the empirical objective converges to the optimizer of the objective as the number of bags grows. Theorem 2 and 3 essentially prove that the proposed algorithm indeed solves the LLP problem under the posed assumptions.

3. The theoretical framework is mapped to an implementable algorithm by an efficient computational strategy to identify some coefficients. Empirical evidence suggests that the proposed method outperforms several baselines on the benchmark datasets.

**Weaknesses:**

1. This work assumes all bags are i.i.d.. I don't think this is a realistic approximation of real-world problems. For example, in the election polling application in the introduction, I think the label proportions of two voting districts in the same city can be highly correlated.

2. It is not clear to me how over-smoothing is mitigated by Theorem 1. Is there a 1-1 relationship between group-level and instance-level outputs, held for other proportion matching methods?

3. The bound (17) seems counterintuitive to me due to the presence of m in the numerator. If you have a fixed number of bags but with an increasing number of instances in each bag, the bound does not get tighter. This is basically saying you're not getting better convergence with more data allocated into a fixed number of bags. Does this reveal certain caveats of the proposed method?

4. Only synthetic datasets are used in the experiments. While many LLP papers used to run experiments on synthetic datasets due to the lack of benchmarks, I think the situation is different with the new LLP benchmarks published in [1].

5. The bag size used in experiments is too small to demonstrate the scalability of the proposed algorithm. The largest bag size in [1] is 512. Papers like [2] even push the largest bag size to 2048. In the tables of [2], the accuracy of some methods rapidly drops after the bag size goes over a certain threshold; I'm wondering if this could also happen to the proposed algorithm.

6. It looks like all instances in the same bag must be pushed to the GPU simultaneously to compute the loss and backpropagate. This seems to be a huge computational disadvantage, especially if the bag sizes are large.

## References:

[1]Brahmbhatt, A., Pokala, M., Saket, R., & Raghuveer, A. (2024). LLP-Bench: A large scale tabular benchmark for learning from label proportions. In Proceedings of the 33rd ACM International Conference on Information and Knowledge Management (CIKM '24) (pp. 4374–4381). Association for Computing Machinery. https://doi.org/10.1145/3627673.3680032

[2] Zhang, J., Wang, Y., & Scott, C. (2022). Learning from label proportions by learning with label noise. In A. H. Oh, A. Agarwal, D. Belgrave, & K. Cho (Eds.), Advances in Neural Information Processing Systems. https://openreview.net/forum?id=cqyBfRwOTm1

**Questions:**

1. How important is it for different bags to have the same size m? I don't think Theorems 2 and 3 depend on a fixed m across all bags (Theorem 3 will probably have a different form due to different bag sizes). Can the authors confirm my observations?

2. While Theorem 2 is acceptable, I'm wondering if an excess risk bound like the styles in [1] can be developed. An excess risk bound could illustrate the sensitivity between $f_k$ and $g_k$ in Theorem 2. To be precise, if we obtained a $g_k$ that is not equal but very close to $g_k^\*$, would $f_k$ also stay close to $f_k^\*$?

3. How would the accuracy look if you fix the number of bags and keep increasing the bag sizes? If (17) is tight, you should not get better convergence.

4. See other questions in weaknesses.

I'm open to increasing the score if some of the concerns and questions can be addressed.


## References:

[1] Steinwart, I. How to Compare Different Loss Functions and Their Risks. Constr Approx 26, 225–287 (2007). https://doi.org/10.1007/s00365-006-0662-3

---

> ### Author Response · Authors · 2025-11-22
>
> We sincerely appreciate your time and effort in reviewing our paper.
>
> **Response to W1: In Some Real-World Scenarios, Bags May Be Non-i.i.d.**
>
> The generation of i.i.d. bags has been widely used in privacy-preserving settings[1], with important applications, for example, in ICML 2025[2], NeurIPS 2023[3], and ICLR 2025[4]. In future work, we plan to extend our framework to non-i.i.d. settings.
>
> **Response to W2: Further Explanation of Theorem 2**
>
> Theorem 2 shows that over-smoothing is effectively mitigated. The detailed reasoning is provided in the proof around line 889. In simple terms, Theorem 1 establishes that the posterior probability of the proportional value is in one-to-one correspondence with a set of the instance-level posterior probability. Under the deterministic setting, the posterior of the proportional value becomes a one-hot vector, and the corresponding instance-label posterior is also one-hot (i.e., of the form 0–1). Therefore, the minimizer of our proposed risk (Eq. 8) is able to avoid over-smoothing.To our knowledge, no other proportional matching method has considered this one-to-one relationship between instance- and bag-level model outputs.
>
> **Response to W4: Without Using Existing LLP Benchmarks**
>
> The main contribution of [1] lies in its feature-bag construction, where feature-similar instances are clustered and then grouped into bags. In our work, we primarily focus on the random-bag setting, which plays an important role in privacy-preserving scenarios. This generation mechanism has been widely adopted in many existing studies that follow this setting[2,3,4]. Moreover, [1] is specifically designed for binary classification and is constructed from a fully supervised dataset, whereas our study addresses the multi-class LLP setting. Because the dataset in [1] is very large (11.15 GB), we plan to include a comparison with this benchmark in the final version of the paper. In the current version, we instead supplement our study with experiments on bags constructed using a feature-similar strategy to demonstrate the effectiveness of our method: we first apply k-means clustering to obtain a specified number of clusters, and then construct bags from these clusters while constraining the bag size to be between 16 and 256. Please refer to our response to Reviewer QG4B for more implementation details.
>
> The experimental results are shown in the following table.
>
> Since DSQ and EasyLLP are based on fixed bag sizes and cannot be directly scaled to this setting, we do not include them in the comparison.
>
> ### CIFAR10
> |CIFAR10|3125|1562|781|390|
> |-|-|-|-|-|
> | PM | 64.31 ± 1.27 | 68.38 ± 1.45 | 68.00 ± 0.83 | 66.74 ± 1.92 |
> | ROT | 72.18 ± 0.56 | 75.05 ± 0.37 | 75.22 ± 0.78 | 74.58 ± 1.13 |
> | LLP-PVC | **75.96 ± 0.42** | **78.26 ± 0.67** | **77.43 ± 0.95** | **77.19 ± 0.24** |
>
> ### FashionMNIST
> |FashionMNIST|3750|1875| 937 | 468 |
> |-|-|-| - | - |
> | PM |81.86 ± 1.08 | 83.58 ± 0.71 | 82.82 ± 1.89 | 81.72 ± 1.05 |
> | ROT | 85.42 ± 0.59 | 85.89 ± 0.76 | 84.71 ± 1.32 | 83.71 ± 0.68 |
> | LLP-PVC | **88.03 ± 1021** | **87.74 ± 0.84** | **86.34 ± 0.47** | **85.41 ± 0.93** |
>
> The horizontal axis represents the number of generated clusters, and our method consistently achieves the best performance.
>
> **Response to W5: Experiments on Larger Bag Size**
>
> We conducted extensive experiments and further increased the bag size to 1024; see our response to Reviewer QG4B for details. In prior work under the random-bag generation scheme for multiclass problems, the largest commonly used bag sizes are typically 128 or 256, which is why we did not initially consider much larger bags. Our results show that, in the multiclass setting, performance with a bag size of 1024 becomes close to random guessing. By contrast, some previous binary classification studies can still learn effectively with bag sizes up to 2048 due to the simpler nature of the task, or they adopt alternative bag-generation schemes (e.g., [8]) where each bag contains a large number of instances from one or only a few classes, which is considerably easier than fully random bag construction. For experimental results and further discussion, please see our response to Reviewer QG4B’s Question 2 and Reviewer cMjG’s Weakness 1.
>
> [1] Google. Private aggregation api of chrome privacy sandbox. https://developer.chrome.com/docs/privacy-sandbox/aggregation-service/, 2024.
>
> [2] Nearly Optimal Sample Complexity for Learning with Label Proportions. ICML 2025
>
> [3] Easy Learning from Label Proportions. NeurIPS 2023
>
> [4] Learning from Aggregate Responses: Instance-Level versus Bag-Level Loss Functions. ICLR 2024
>
> [5] https://lightning.ai/docs/pytorch/stable/common/gradient_accumulation.html
>
> [6] Making Deep Neural Networks Robust to Label Noise: a Loss Correction Approach. CVPR 2017
>
> [7] Consistent Multi-Class Classification from Multiple Unlabeled Datasets. ICLR 2024
>
> [8] Zhang, J., Wang, Y., & Scott, C. (2022). Learning from label proportions by learning with label noise.

---

> ### Author Response · Authors · 2025-11-25
>
> **Response to W6: A bag that is too large may not fit into GPU memory.**
>
> This issue can be effectively addressed by standard gradient accumulation: a large bag is split into several micro-batches that are processed sequentially, with their gradients accumulated before a single parameter update.  This strategy is already supported by mature tooling in the PyTorch ecosystem [5].
>
> **Response to Q1 and 3: On the Necessity of a Fixed Bag Size in Theory and Experiments**
>
> For Theorem 1, Theorem 2, and the experimental section, our results do not rely on the fixed bag size m. In the experiments, we use a fixed bag size simply to illustrate the effectiveness of our method under different supervision levels.
>
> For Theorem 3, we characterize the error bound. This bound increases as the bag size grows. The reason is twofold: it is related to our $m$-enumerator construction, and it also stems from the fact that $\mathcal{L}\left(\mathbf{g}_k(\mathbf{B}), \tilde{\alpha}_k\right)$ increases with $m$; see Lemma 5 for details. Using a fixed bag size $m$ makes this relationship easier to characterize and leads to a more transparent analysis. Without fixing $m$, one would need to introduce additional assumptions about the distribution from which the bag size is drawn, which makes no essential difference. In existing works that provide similar bounds, the bag size also appears in the numerator of the bound [2,3].
>
> **Response to W3: Effects of Increasing Bag Size (with a Fixed Number of Bags) on Performance and Bound Tightness**
>
> We conduct an additional experiment on FMNIST and do not observe any clear improvement when increasing the bag size while keeping the number of bags fixed. We believe this only provides partial evidence that our generalization bound is reasonably tight, since the empirical results can also be influenced by other factors—for example, smaller bag sizes may lead to overfitting issues when training neural networks.
>
> In LLP, using larger bags actually provides weaker supervision, because the label information becomes more coarse-grained at the instance level. Simply enlarging the bag size is therefore not inherently beneficial: although the total number of instances increases, the effective supervision per instance decreases. To the best of our knowledge, in all existing LLP methods the theoretical bounds explicitly depend on the bag size in the numerator[2,3]; as the bag size grows, these bounds become looser.
>
> |16|32|64|128|
> |-|-|-|-|
> | 86.92 ± 0.41 | 86.49 ± 0.36 | 87.03 ± 0.57 | 87.02 ± 0.24 |
>
> **Response to Question 2: Can the Excess Risk Bound Illustrate the Sensitivity Between $g\_k$ and $f\_j$ in Theorem 2?**
>
> Our theoretical results only ensure that, as the number of bags $n \to \infty$, the learned $g_k$ converges to $g_k^\star$. From this, we can derive a relationship between $f_k^\star$ and $g_k^\star$; however, to the best of our knowledge, we are not yet able to show that when $g_k$ is not exactly equal to but only close to $g_k^\star$, the corresponding $f_k$ will also stay close to $f_k^\star$. Arguments of this form are in fact widely used in weakly supervised learning, for example, in UU learning [7] and in label-noise learning with the classical forward correction [2].
>
> Obtaining an excess risk bound that explicitly characterizes the sensitivity between $f_k$ and $g_k$ typically requires much stronger theoretical guarantees, such as when the newly proposed loss is an unbiased risk estimator (URE) of $\bar{R}_k(f_k)$ in (4) or of the risk $R(f)$ in (2). In practice, however, URE-based methods come with stronger theoretical guarantees, but in practice they often suffer from the following drawbacks: for instance, EasyLLP [3] may produce negative empirical risks, which can lead to overfitting.

---

### Official Review · Reviewer_QG4B · 2025-10-27

**Soundness:** 2
**Presentation:** 3
**Contribution:** 3
**Rating:** 4
**Confidence:** 4

**Summary:**

This paper proposes a novel approach for Learning from Label Proportions (LLP) that may mitigate the over-smoothing problem common in proportion matching strategies with theoretical guarantees. The core contribution is to treat label proportions as labels for an auxiliary proportional value classification (PVC) task, which is then used to induce the instance-level classifier. The authors also introduce a "divide-and-conquer" computational method using FFT to solve the problem with an efficient time complexity. Experiments on text and image datasets show that the method achieves better performance against other LLP methods in settings with random bags with fixed size.

**Strengths:**

- **Novelty**: The core idea to solve a different, auxiliary problem, PVC, is creative and new for this field.
- **Efficiency**: The paper introduces a clever and practical solution using the FFT to make the computationally complex aggregation step efficient.
- **Theoretical quality**: The paper provides strong theoretical backing for the PVC approach. It includes formal guarantees showing how the method mitigates over-smoothing and connects the bag-level task back to the instance-level classifiers.
- **Clarity**: The paper is well-written and clear

**Weaknesses:**

1. **Mismatch between motivation and experimental setup**: The authors motivate the LLP problem with real-world applications like election polling (Section 1), which often involve complex, non-random bag generation (see [1] for a real-world application of LLP in election). However, the experimental setup (Section 4.1) relies exclusively on bags generated randomly with a fixed size. This setup only addresses one variant of LLP (termed "Naive" in the taxonomy proposed by [2]) and is unlikely to represent the more complex scenarios used as motivation. Even though using real-world data such as [1] is unfeasible, other works have explored more realistic bag creation methods, such as or clustering [3] to create dependence structures that are more realistic. The paper's claims would be significantly strengthened by testing against these more challenging and representative settings.

2. **Limited bag size exploration**: Even within the chosen random setting, it is known that LLP performance often degrades as bag size increases. The bag sizes tested in the paper are relatively small (a maximum of 128 for text and 32 for images, per Section 4.1). The claims about the method's robustness would be better supported by an analysis of its performance with much larger bag sizes, as seen in other recent work (e.g., up to 2048 in [5]).

3. **Missing baseline comparisons**: The evaluation is missing comparisons against several important and recent baselines, such as [4], [5], [6], and [7]. These methods are highly relevant to the task, and their omission makes it difficult to assess the proposed method's performance in the context of the current state-of-the-art. Notably, [5] is a recent ICLR 2024 publication, making its absence in the comparison tables particularly significant.

4. **Potentially unfair hyperparameter selection**: The paper states that "all methods shared the same hyperparameter settings" (Section 4.1) to ensure fairness. While using the same network architecture is reasonable, fixing hyperparameters like the learning rate (LR) for all methods (detailed in Tables 5 and 6) can unfairly disadvantage baselines, as different optimization methods often have different optimal LRs. A more robust comparison would involve minimal tuning (e.g., testing a small set of LR values) for each baseline and reporting its best performance. This would provide a stronger guarantee that the proposed method's superiority is not an artifact of a single, fixed hyperparameter choice that favors it over others.

**Minor details**:
- $m$ is used in line 98 before its definition (line 142).
- Figure 1 labels are a bit confusing. I would suggest to have entropy and accuracy as two different colors, and have different methods with different markers.


[1]: Assessing Candidate Preference through Web Browsing History: https://dl.acm.org/doi/pdf/10.1145/3219819.3219884
[2]: Dependence and Model Selection in LLP: The Problem of Variants: https://dl.acm.org/doi/pdf/10.1145/3580305.3599307
[3]: (Almost) no label no cry: https://proceedings.neurips.cc/paper_files/paper/2014/file/d3313de3f431fd64513431c4326d237c-Paper.pdf
[4]: Learning from Label Proportions with Consistency Regularization: https://proceedings.mlr.press/v129/tsai20a/tsai20a.pdf
[5]: Learning from Label Proportions: Bootstrapping Supervised Learners via Belief Propagation: https://openreview.net/pdf?id=KQe9tHd0k8
[6]: MixBag: Bag-Level Data Augmentation for Learning from Label Proportions: https://openaccess.thecvf.com/content/ICCV2023/papers/Asanomi_MixBag_Bag-Level_Data_Augmentation_for_Learning_from_Label_Proportions_ICCV_2023_paper.pdf
[7]: Learning from label proportions by learning with label noise: https://proceedings.neurips.cc/paper_files/paper/2022/file/ac56fb3fab015124b541f6299016a21c-Paper-Conference.pdf

**Questions:**

1. Could you clarify the intended scope of LLP-PVC? The motivation (Section 1) suggests broad use, but the experiments (Section 4.1) use only random fixed-size bags. Do you expect this performance to generalize to non-random settings (e.g., clustered bags [3])?
2. Why were experiments limited to small bag sizes (max 128)? Was there another bottleneck? Can you provide insight into the expected performance as bag size grows significantly?  From the literature, we expect the performance to decrease with the bag size increase.
3. Could you provide a brief conceptual comparison to recent baselines [4, 5, 6, 7], especially [5] (ICLR 2024)? What are the key methodological differences and trade-offs versus [5]?
4. Can you justify using a single fixed LR for all methods? This choice may unfairly disadvantage baselines. Is there evidence this choice doesn't create a confound, or that this LR is a reasonable default for all?

---

> ### Author Response · Authors · 2025-11-22
>
> We sincerely appreciate your time and effort in reviewing our paper.
>
> **Q1: Experiments Do Not Cover Some Bag-Generation Scenarios**
>
> Our work focuses on the standard random-bag LLP setting, which has mainly been explored in privacy-preserving scenarios [9], partly explaining the scarcity of real-world public datasets, and has recently attracted substantial attention with many works adopting this formulation [10] (ICML 2025) and [12] (ICLR 2024). Constructing bags via similarity-based procedures would tend to group feature- and label-similar samples, thereby increasing the risk of privacy leakage and targeted attacks, whereas randomly formed bags do not suffer from this issue and may even offer stronger protection in sensitive applications. Both similarity-based and random bag constructions are practically relevant settings in LLP.
>
> **LLP-PVC can be readily applied to more general bag-generation schemes**. If the bags B are i.i.d. and the instance labels are conditionally independent given their features, then our method and all theoretical results continue to hold in this setting. Importantly, this remains true even when the instance features within each bag exhibit arbitrary dependence structures. For example, the instances can be induced by graphical models or Markov chains, or they can be generated by first sampling proportions in a certain manner and then drawing the instances accordingly, as in [11]. Most theoretical analyses are only applicable under the assumption that the instances within each bag are i.i.d [10,12]. When the bag construction procedure violates the i.i.d. bag assumption (e.g., the training set is partitioned once by clustering), LLP-PVC can still be applied as a heuristic under the conditional independence assumption, although our theoretical guarantees no longer strictly apply. To the best of our knowledge, existing methods in this non-i.i.d. bag setting are purely heuristic as well.
>
> On the other hand, our method can be readily extended to the case where the bag size is not fixed. However, since the bag size in LLP directly reflects the strength of supervision, using fixed bag sizes allows us to more clearly and transparently evaluate how our method behaves under different supervision levels.
>
> **Q2: Experiments on Larger Bag Size**
>
> Our bag-size setting follows prior multi-class LLP works that randomly group instances into bags, e.g., [4,13], where the maximum bag size is 256 or 128, we extended the image-dataset experiments to larger bag sizes; please see our response to Reviewer cMjG. For binary LLP, as in [5], one can often use much larger bags and still learn effectively, since binary classification is relatively simple; however, this does not hold in the multi-class case. As shown in our KMNIST results, when the bag size reaches 512 the performance already degrades significantly, and at 1024 it is close to random, so further increasing the bag size is not meaningful. In contrast, [7] uses much larger bags and constructs them in an uncommon way by enforcing different class priors across bags. This results in highly diverse and unbalanced label proportions, unlike the nearly identical proportions produced by random grouping around the global class prior, which makes learning more difficult. In addition, [7] periodically re-forms the bags during training (every 20 epochs, as stated in its appendix), all of the above allows it to support very large bag sizes.
>
> **Q3: Miss Baseline Comparision**
>
> In [4], the authors introduce a consistency regularization term combined with a proportion-matching strategy to improve empirical performance, we mainly compare against vanilla algorithms without regularization. Our method can readily incorporate such a term; as shown in our response to Reviewer cMjG‘s W4, adding a simple consistency-regularization component already brings the performance close to the current SOTA. [6] does not propose a concrete learning algorithm, but rather a bag-level data augmentation scheme; LLP-PVC can also be trained with this type of augmentation.
>
> The method in [5] is explicitly designed for the binary classification setting, as stated in its abstract. It is a heuristic algorithm for obtaining pseudo-labels. Subsequently, these pseudo-labels guide the learning of the embedding. This is fundamentally different from our approach, which directly performs classification using proportional value. For Question 3 in [7], the experimental settings are quite special and differ from ours, so we do not include a direct comparison.
>
> [9] Google. Private aggregation api. https://developer.chrome.com/docs/privacy-sandbox/aggregation-service/ .
>
> [10] Nearly Optimal Sample Complexity for Learning with Label Proportions.
>
> [11] Learning from Label Proportions: A Mutual Contamination Framework
>
> [12] Learning from Aggregate Responses: Instance-Level versus Bag-Level Loss Functions.
>
> [13] Learning from Label Proportions with Generative Adversarial Networks.

---

> ### Author Response · Authors · 2025-11-24
> **Additional Experimental Results for Large Bag Sizes (for Question 2)**
>
> We further provide additional experimental results on large bag sizes and on the more challenging CIFAR-100 dataset.
>
> The experimental settings are identical to those described in the main paper, and the results are summarized in the tables below:
>
> **CIFAR-10**
>
> |Method|64|128|256|512|1024|
> |-|-|-|-|-|-|
> |PM|60.47±0.20|49.21±0.38|36.54±1.53|21.63±3.37|10.26±1.99|
> |DSQ|46.89±0.16|42.19±0.64|41.84±1.74|**31.06±2.11**|10.09±1.58|
> |EasyLLP|44.19±0.94|37.36±1.87|31.57±1.93|23.23±1.90|11.00±0.77|
> |EasyLLP-flood|54.88±0.44|47.54±0.41|39.67±2.08|27.97±0.78|11.00±0.77|
> |ROT|69.59±1.28|33.70±8.02|20.94±4.01|17.30±3.20|**13.63±1.49**|
> |LLP-PVC|**78.90±0.44**|**73.02±1.97**|**42.12±3.32**|22.17±1.69|10.78±1.28|
>
> **FashionMNIST**
>
> |Method|64|128|256|512|1024|
> |-|-|-|-|-|-|
> |PM|75.81±0.68|66.78±5.29|47.86±8.12|30.00±6.15|18.54±1.78|
> |DSQ|77.14±0.42|75.82±0.82|75.04±0.24|71.70±0.71|11.94±2.12|
> |EasyLLP|68.13±0.39|66.27±0.84|64.12±1.51|65.93±1.45|14.64±0.70|
> |EasyLLP-flood|79.22±0.37|73.69±0.35|71.04±0.81|68.51±2.00|14.64±0.70|
> |ROT|81.42±0.34|65.38±5.20|27.59±10.46|18.62±9.67|17.46±5.77|
> |LLP-PVC|**89.28±0.02**|**87.17±0.16**|**79.42±6.33**|**75.40±2.63**|**20.47±3.61**|
>
> **KMNIST**
>
> |Method|64|128|256|512|1024|
> |-|-|-|-|-|-|
> |PM|51.05±4.22|28.30±1.29|23.93±6.40|16.73±3.97|12.55±2.97|
> |DSQ|57.58±2.52|54.90±2.41|51.98±1.86|38.15±10.62|11.00±1.84|
> |EasyLLP|50.60±1.46|48.56±2.84|43.14±3.09|46.03±2.47|12.73±1.81|
> |EasyLLP-flood|70.48±2.13|63.87±1.26|57.03±2.10|50.20±0.61|12.73±1.81|
> |ROT|73.00±4.23|43.37±2.37|16.89±5.04|15.07±0.92|**15.64±2.21**|
> |LLP-PVC|**96.40±0.15**|**96.36±0.16**|**94.95±0.32**|**51.99±8.22**|14.68±7.16|
>
> **CIFAR100**:
>
> |Method|16|32|64|128|256|512|1024|
> |-|-|-|-|-|-|-|-|
> |PM|**57.47 ± 0.63**|53.48 ± 0.41|46.80 ± 0.97|33.48 ± 1.29|13.80 ± 1.27|5.19 ± 0.83|1.82 ± 0.45|
> |DSQ|16.18 ± 0.52|16.13 ± 0.33|15.42 ± 0.61|11.35 ± 0.47|7.63 ± 1.00|2.23 ± 1.08|1.15 ± 0.71|
> |ROT|42.99 ± 0.79|31.07 ± 1.62|14.45 ± 2.51|5.36 ± 1.44|2.98 ± 1.39|1.48 ± 0.95|1.84 ± 0.27|
> |EasyLLP|23.35 ± 2.69|10.72 ± 1.58|4.65 ± 1.44|2.12 ± 2.27|1.56 ± 1.05|2.02 ± 0.47|1.23 ± 0.68|
> |EasyLLP_flood|29.18 ± 0.71|19.66 ± 0.39|8.24 ± 0.65|4.87 ± 0.53|2.87 ± 0.94|1.91 ± 0.32|1.36 ± 0.39|
> |LLP-PVC|57.44 ± 0.55|**55.07 ± 0.63**|**50.55 ± 0.48**|**43.63 ± 0.36**|**19.21 ± 1.73**|**6.89 ± 0.97**|**1.84 ± 0.30**|
>
> We found that our method performs well with bag sizes of 256 and 512. However, when the bag size reaches 1024, almost all methods perform poorly—only slightly better than random. This is because randomly generated bags tend to approximate the prior class distribution: the larger the bag, the closer its class proportions are to the prior distribution. In particular, when the dataset is relatively balanced across classes, this makes the learning task significantly more challenging.
>
> In binary classification, which is relatively simple, models can still learn under this setting [5]. But in multi-class classification, the performance drops dramatically and becomes only marginally better than random. Therefore, previous works that generate bags randomly typically set the maximum bag size to 128 or 256 [4,13,14,15,16,17].
>
> In [7], the bag size is also set to be very large because their data-generation assumption draws samples in each bag from different prior distributions. This assumption makes the class proportions within each bag extremely imbalanced, and in some cases one class may dominate a bag, which in fact makes the learning problem easier. Their experimental setup describes this issue, and the authors also discuss the lack of real-world plausibility of this assumption in the limitations section of [7].
>
> [14] Forming Auxiliary High-confident Instance-level Loss to Promote Learning from Label Proportions. CVPR2025
>
> [15] Two-stage Training for Learning from Label Proportions. IJCAI 2021
>
> [16] A Two-Stage Training Framework with Feature-Label Matching Mechanism for Learning from Label Proportions. ACML 2021
>
> [17] Learning from Label Proportions with Prototypical Contrastive Clustering. AAAI 2022

---

> ### Author Response · Authors · 2025-11-29
> **Experiments with Different Learning Rates (Question 4)**
>
> **Response to Q4**
>
> The table below reports the results on the AGNews under learning rates of 5e-5, 5e-6 (our default setting), and 5e-7.
>
> lr=5e-5:
>
> |Method|16|32|64|128|
> |-|-|-|-|-|
> |PM|87.29|71.52|66.71|65.70|
> |DSQ|87.66|72.15|66.08|66.60|
> |EasyLLP|88.04|85.08|82.58|66.61|
> |EasyLLP-flood|87.81|87.12|83.01|66.61|
> |ROT|87.25|68.90|66.72|66.12|
> |LLP-PVC|**90.49**|**89.33**|**87.75**|**87.02**|
>
> lr=5e-6
>
> |Method|16|32|64|128|
> |-|-|-|-|-|
> |PM|87.37|83.58|75.94|67.43|
> |DSQ|87.08|83.38|77.14|67.14|
> |EasyLLP|88.13|86.67|84.77|64.81|
> |EasyLLP-flood|88.35|86.95|84.77|64.82|
> |ROT|86.94|83.79|78.56|69.63|
> |LLP-PVC|**91.16**|**89.82**|**89.09**|**87.81**|
>
> lr=5e-7
>
> |Method|16|32|64|128|
> |-|-|-|-|-|
> |PM|80.32|68.36|51.84|41.33|
> |DSQ|80.75|68.56|52.13|41.42|
> |EasyLLP|85.48|80.99|70.34|41.34|
> |EasyLLP-flood|85.48|80.93|69.67|41.34|
> |ROT|80.29|68.40|51.86|41.34|
> |LLP-PVC|**85.58**|**82.15**|**71.93**|**58.36**|
>
> We also tested learning rates 5e-4 and 5e-8, which led to uniformly poor performance for all methods, and thus are not reported. We consider 5e-6 to be an approximately optimal and fair learning rate for all methods.

---

> ### Author Response · Authors · 2025-11-29
> **Additional Experimental Results for Bag Construction Using Clustering (Question 1)**
>
> We treat LLP-PVC as a heuristic algorithm and extend it to the clustering setting. Following [4], we use clustering to generate bags. First, we generate a certain number of clusters using the K-means algorithm. Then, we directly take the clusters whose sizes fall between 16 and 256 as bags. For the remaining clusters, those larger than 256 are gradually split into bags with a maximum size of 256. The leftover clusters and those smaller than the minimum bag size of 16 are combined into a new large cluster, which is then further split into smaller bags. All other experimental settings follow the original paper.
>
> Since DSQ and EasyLLP are based on fixed bag sizes and cannot be directly scaled to this setting, we do not include them in the comparison.
>
>
> ### CIFAR10
> | CIFAR10 | 3125 | 1562 | 781 | 390 |
> | - | - | - | - | - |
> | PM | 64.31 ± 1.27 | 68.38 ± 1.45 | 68.00 ± 0.83 | 66.74 ± 1.92 |
> | ROT | 72.18 ± 0.56 | 75.05 ± 0.37 | 75.22 ± 0.78 | 74.58 ± 1.13 |
> | LLP-PVC | **75.96 ± 0.42** | **78.26 ± 0.67** | **77.43 ± 0.95** | **77.19 ± 0.24** |
>
> ### FashionMNIST
> | FashionMNIST | 3750 | 1875 | 937 | 468 |
> | - | - | - | - | - |
> | PM | 81.86 ± 1.08 | 83.58 ± 0.71 | 82.82 ± 1.89 | 81.72 ± 1.05 |
> | ROT | 85.42 ± 0.59 | 85.89 ± 0.76 | 84.71 ± 1.32 | 83.71 ± 0.68 |
> | LLP-PVC | **88.03 ± 1021** | **87.74 ± 0.84** | **86.34 ± 0.47** | **85.41 ± 0.93** |
>
> The horizontal axis represents the number of generated clusters, and our method consistently achieves the best performance.

---

> ### Author Response · Authors · 2025-12-02
> **Additional Experiments for Scenarios Where Instances Within a Bag Are Not Independent (Question 1)**
>
> We also designed an experimental setting where instances within each bag exhibit dependency. Specifically, we first generate the target label proportion and then collect instances according to this proportion to construct the bag. To construct each bag, we first draw a class-proportion vector $\alpha$ from the uniform distribution over the probability simplex $\Delta\_q$ (equivalently, $(\alpha \sim \text{Dirichlet}(1,\ldots,1))$.
> Given $\alpha$ and the desired bag size, we then sample the number of instances per class from a multinomial distribution with parameter $\alpha$ and select that many data points without replacement from each class to form the bag.
>
> ### FashionMNIST
>
> | Method   | 16              | 32              | 64              | 128             |
> |---------|-----------------|-----------------|-----------------|-----------------|
> | PM      | 87.21 ± 1.42    | 82.42 ± 1.18    | 76.62 ± 0.96    | 71.34 ± 1.73    |
> | ROT     | 91.31 ± 0.67    | 87.51 ± 1.54    | 84.10 ± 1.25    | 79.91 ± 1.07    |
> | **LLP-PVC** | **96.32 ± 0.23** | **96.47 ± 0.32** | **96.30 ± 0.39** | **94.20 ± 0.75** |
>
> ---
>
> ### KMNIST
>
> | Method   | 16              | 32              | 64              | 128             |
> |---------|-----------------|-----------------|-----------------|-----------------|
> | PM      | 85.48 ± 1.57    | 82.85 ± 0.89    | 84.04 ± 1.23    | 79.79 ± 1.81    |
> | ROT     | 87.39 ± 1.12    | 85.91 ± 1.68    | 84.04 ± 1.05    | 82.16 ± 0.63    |
> | **LLP-PVC** | **91.30 ± 0.92** | **90.96 ± 0.37** | **90.16 ± 0.74** | **88.55 ± 1.26** |
>
> ---
>
> ### CIFAR10
>
> | Method   | 16              | 32              | 64              | 128             |
> |---------|-----------------|-----------------|-----------------|-----------------|
> | PM      | 74.32 ± 1.28    | 71.08 ± 1.53    | 69.38 ± 0.82    | 67.56 ± 1.47    |
> | ROT     | 76.83 ± 1.65    | 72.86 ± 1.19    | 71.44 ± 0.97    | 70.72 ± 1.33    |
> | **LLP-PVC** | **80.78 ± 0.78** | **80.29 ± 0.42** | **79.54 ± 1.01** | **79.70 ± 1.09** |
>
> We did not include comparisons with EasyLLP or the DSQ baseline, as both methods are explicitly designed under the assumption that instances within each bag are i.i.d. Nevertheless, our results show that under this dependent-instance setting, LLP-PVC still significantly outperforms the other baselines.

---

### Official Review · Reviewer_cMjG · 2025-10-28

**Soundness:** 3
**Presentation:** 2
**Contribution:** 2
**Rating:** 4
**Confidence:** 4

**Summary:**

This paper proposed a novel LLP approach that can mitigate the longstanding over-smoothing problem with only the incorporation of an aggregation function after the classification layer. An efficient computational approach with a divide-and-conquer strategy was also proposed.

**Strengths:**

1.	The proposed method is simple to implement, only involves with an auxiliary proportional value classification task.
2.	A divide-and-conquer method is adopted to reduce the computational complexity by using the fast Fourier transform, which largely reduces the computational cost.
3.	The theoretical work guarantees that the model outputs can mitigate over-smoothing problems under certain assumptions. It also analyses the convergence rate of the proposed risk estimator by providing an estimation error bound.

**Weaknesses:**

1.	The datasets and bag sizes for the image classification in the empirical study seems to be small. The authors should consider larger datasets with larger bag sizes, for example, CIFAR-100 and bag sizes of 32, 64, and 128.
2.	The actual running time reported in Table 3 is not consistent to the time complexity results.
3.	The reported results seem to be inferior to that is reported in former work. For example, CIFAR-10 in Table 2.
4.	Most of the compared methods are not state-of-the-arts.
5.	Conclusion is too weak and there is no limitation indicated.

**Questions:**

1.	What is the reason of the results with larger bag sizes are higher than that of smaller bag sizes for the proposed method, for example, results on K-MNIST in Table 2.
2.	Is there any hyperparamters sensitivity and ablation study should be investigated?

---

> ### Author Response · Authors · 2025-11-22
> **Response to Weakness 1-2**
>
> We sincerely appreciate your effort in reviewing our paper.
>
> **Response to Weakness 1 – Experiments on larger bags and more complex datasets**
>
> We further conduct experiments on the MNIST series and CIFAR-10 with bag sizes up to 1024, as well as additional experiments on CIFAR-100, while keeping all other experimental settings the same as in the original setup.
>  The corresponding results will be presented in the final paper.
>
> | Dataset / Method |64|128| 256| 512| 1024|
> |-|-|-|-|-|-|
> | **CIFAR-10**     |                  |                  |                  |                  |                  |
> | PM | 60.47 ± 0.20| 49.21 ± 0.38     | 36.54 ± 1.53     | 21.63 ± 3.37     | 10.26 ± 1.99     |
> | DSQ              | 46.89 ± 0.16     | 42.19 ± 0.64     | 41.84 ± 1.74     | **31.06 ± 2.11** | 10.09 ± 1.58     |
> | EasyLLP          | 44.19 ± 0.94     | 37.36 ± 1.87     | 31.57 ± 1.93     | 23.23 ± 1.90     | 11.00 ± 0.77     |
> | EasyLLP-flood    | 54.88 ± 0.44     | 47.54 ± 0.41     | 39.67 ± 2.08     | 27.97 ± 0.78     | 11.00 ± 0.77     |
> | ROT              | 69.59 ± 1.28     | 33.70 ± 8.02     | 20.94 ± 4.01     | 17.30 ± 3.20     | **13.63 ± 1.49** |
> | LLP-PVC          | **78.90 ± 0.44** | **73.02 ± 1.97** | **42.12 ± 3.32** | 22.17 ± 1.69     | 10.78 ± 1.28     |
> | **FashionMNIST** |                  |                  |                  |                  |                  |
> | PM               | 75.81 ± 0.68     | 66.78 ± 5.29     | 47.86 ± 8.12     | 30.00 ± 6.15     | 18.54 ± 1.78     |
> | DSQ              | 77.14 ± 0.42     | 75.82 ± 0.82     | 75.04 ± 0.24     | 71.70 ± 0.71     | 11.94 ± 2.12     |
> | EasyLLP          | 68.13 ± 0.39     | 66.27 ± 0.84     | 64.12 ± 1.51     | 65.93 ± 1.45     | 14.64 ± 0.70     |
> | EasyLLP-flood    | 79.22 ± 0.37     | 73.69 ± 0.35     | 71.04 ± 0.81     | 68.51 ± 2.00     | 14.64 ± 0.70     |
> | ROT              | 81.42 ± 0.34     | 65.38 ± 5.20     | 27.59 ± 10.46    | 18.62 ± 9.67     | 17.46 ± 5.77     |
> | LLP-PVC          | **89.28 ± 0.02** | **87.17 ± 0.16** | **79.42 ± 6.33** | **75.40 ± 2.63** | **20.47 ± 3.61** |
> | **KMNIST**       |                  |                  |                  |                  |                  |
> | PM               | 51.05 ± 4.22     | 28.30 ± 1.29     | 23.93 ± 6.40     | 16.73 ± 3.97     | 12.55 ± 2.97     |
> | DSQ              | 57.58 ± 2.52     | 54.90 ± 2.41     | 51.98 ± 1.86     | 38.15 ± 10.62    | 11.00 ± 1.84     |
> | EasyLLP          | 50.60 ± 1.46     | 48.56 ± 2.84     | 43.14 ± 3.09     | 46.03 ± 2.47     | 12.73 ± 1.81     |
> | EasyLLP-flood    | 70.48 ± 2.13     | 63.87 ± 1.26     | 57.03 ± 2.10     | 50.20 ± 0.61     | 12.73 ± 1.81     |
> | ROT              | 73.00 ± 4.23     | 43.37 ± 2.37     | 16.89 ± 5.04     | 15.07 ± 0.92     | **15.64 ± 2.21** |
> | LLP-PVC          | **96.40 ± 0.15** | **96.36 ± 0.16** | **94.95 ± 0.32** | **51.99 ± 8.22** | 14.68 ± 7.16     |
>
>
> CIFAR100:
>
> |Method|16|32|64|128|256|512|1024|
> |-|-|-|-|-|-|-|-|
> |PM|**57.47 ± 0.63**|53.48 ± 0.41|46.80 ± 0.97|33.48 ± 1.29|13.80 ± 1.27|5.19 ± 0.83|1.82 ± 0.45|
> |DSQ|16.18 ± 0.52|16.13 ± 0.33|15.42 ± 0.61|11.35 ± 0.47|7.63 ± 1.00|2.23 ± 1.08|1.15 ± 0.71|
> |ROT|42.99 ± 0.79|31.07 ± 1.62|14.45 ± 2.51|5.36 ± 1.44|2.98 ± 1.39|1.48 ± 0.95|1.84 ± 0.27|
> |EasyLLP|23.35 ± 2.69|10.72 ± 1.58|4.65 ± 1.44|2.12 ± 2.27|1.56 ± 1.05|2.02 ± 0.47|1.23 ± 0.68|
> |EasyLLP_flood|29.18 ± 0.71|19.66 ± 0.39|8.24 ± 0.65|4.87 ± 0.53|2.87 ± 0.94|1.91 ± 0.32|1.36 ± 0.39|
> |LLP-PVC|57.44 ± 0.55|**55.07 ± 0.63**|**50.55 ± 0.48**|**43.63 ± 0.36**|**19.21 ± 1.73**|**6.89 ± 0.97**|**1.84 ± 0.30**|
>
> **Response to Weakness 2 – Discrepancy between theoretical time complexity and empirical runtime**
>
> The time-complexity analysis characterizes how the running time grows with respect to n in terms of order of magnitude, up to constant factors. In contrast, the actual wall-clock times are affected by many additional factors, such as hardware configuration, implementation details, low-level optimizations, and the constant terms hidden in the big-O notation. As a result, some mismatch between these two measures is expected; they are complementary indicators of the time performance of an algorithm rather than quantities that must coincide exactly. For a more detailed discussion, please see[1].
>
>
> [1] Introduction to Algorithms

---

> ### Author Response · Authors · 2025-11-23
> **Response to Weakness 3-5**
>
> **Response to Weakness 3 – Results seem to be inferior to that is reported in former work**
>
> All experimental results in our paper are obtained from our own re-implementations. The main discrepancy compared with former work comes from the choice of backbone networks.
> For example, while [2,3] employ WRN-28-2, we adopt ResNet-18, which is a lighter-weight backbone and typically yields lower accuracy than WRN-28-2 on CIFAR-10.
>
> **Response to Weakness 4 – Lack of comparison with state-of-the-art  methods**
>
> All compared methods are vanilla weakly supervised baselines without additional regularization or representation-learning components.
>
> We further provide additional experiments comparing our method with state-of-the-art approaches. Specifically, we adopt two types of data augmentation and incorporate a FixMatch-style regularization term on top of LLP-PVC to further enhance model performance. This regularization enforces consistency by using weakly augmented samples whose predicted confidence exceeds a threshold (0.95) to supervise their strongly augmented counterparts. All experimental settings follow those used in L²P-AHIL [2] (CVPR 2025). The other baseline methods are taken directly from the original implementation in [2], and they represent the most competitive approaches currently available.
>
>
> **CIFAR-10**
>
> | Model     | 16               | 32               | 64               | 128              |
> |-----------|------------------|------------------|------------------|------------------|
> | DLLP      | 91.59 ± 1.52     | 88.61 ± 0.90     | 79.76 ± 1.45     | 64.95 ± 0.01     |
> | LLP-VAT   | 91.80 ± 0.08     | 89.11 ± 0.22     | 78.75 ± 0.46     | 63.89 ± 0.19     |
> | ROT       | 94.86 ± 0.68     | 94.34 ± 0.65     | 93.97 ± 0.96     | 92.23 ± 0.81     |
> | SoftMatch | 95.24 ± 0.12     | 95.25 ± 0.14     | 94.23 ± 0.18     | **93.87 ± 0.22** |
> | FLMm*     | 92.34            | 92.00            | 91.74            | 91.54            |
> | L²P-AHIL  | 94.96 ± 0.13     | 95.00 ± 0.11     | 94.58 ± 0.21     | 93.64 ± 0.20     |
> | LLP-PVC   | **95.60 ± 0.21** | **95.34 ± 0.18** | **95.03 ± 0.27** | 91.03 ± 0.09     |
>
>
> **CIFAR-100**
>
> | Model     | 16                | 32                | 64                | 128                |
> |-----------|-------------------|-------------------|-------------------|--------------------|
> | DLLP      | 71.28 ± 1.56      | 69.92 ± 2.86      | 53.58 ± 1.60      | 25.86 ± 2.15       |
> | LLP-VAT   | 73.85 ± 0.22      | 71.62 ± 0.07      | 65.31 ± 0.33      | 37.36 ± 0.63       |
> | ROT       | 72.74 ± 0.08      | 69.31 ± 0.22      | 17.48 ± 0.86      | 11.02 ± 0.79       |
> | SoftMatch | 80.14 ± 0.12      | 2.40 ± 0.15       | 2.04 ± 0.10       | 2.12 ± 0.13        |
> | FLMm*     | 66.16             | 65.59             | 64.07             | 61.25              |
> | L²P-AHIL  | 78.65 ± 0.28      | 77.30 ± 0.50      | 76.52 ± 0.23      | 72.21 ± 0.37       |
> | LLP-PVC   | **81.90 ± 0.17**  | **81.77 ± 0.23**  | **80.37 ± 0.11**  | **76.06 ± 0.29**   |
>
> From these results, we can see that our method attains state-of-the-art performance after simply adding a lightweight regularization term. In the main paper, however, we deliberately focus on comparisons using the “vanilla” version of our method without this regularization term, in order to provide a fair and clear evaluation of the core framework.
>
> In addition, we note that the ROT results in this table are obtained from the implementation in [1], which also employs strong–weak data augmentation as a regularization mechanism, whereas our method in this comparison does not use such regularization. This explains why ROT performs better than our approach in this setting.
>
> **Response to Weakness 5 - Discussion of Conclusion and Limitations**
>
> Our work proposes an LLP method based on proportional value classification, which theoretically addresses the over-smoothing problem and achieves strong empirical performance. A limitation of our approach is that it relies on an i.i.d. bag-generation process, which is the standard assumption in LLP and is commonly adopted in privacy-preserving applications. Extending our method to non-i.i.d. settings is an important direction for future work.
>
>
> [2] Forming Auxiliary High-confident Instance-level Loss to Promote Learning from Label Proportions
>
> [3] Learning from Label Proportions with Consistency Regularization
>
> [4] Easy Learning from Label Proportions

---

> ### Author Response · Authors · 2025-11-29
> **Response to Questions**
>
> **Response to Question 1: Why does the method sometimes perform better on a few large bag sizes than on small bag sizes?**
>
> For example, with bag size 8 we obtain 96.62 ± 0.29, only slightly lower than 96.68 ± 0.22 for bag size 16; this 0.06% gap is well within the standard deviation and is not statistically meaningful, likely due to experimental variance. Moreover, KMNIST is a relatively simple dataset, so the model can still learn well under weaker supervision, whereas on the other datasets we observe a clear and consistent performance degradation as the bag size increases.
>
> **Response to Question 2: Why are there no ablation and parameter sensitivity experiments?**
>
> Note that LLP-PVC does not introduce any additional hyperparameters; it is formulated as a single loss term that is directly used for training, further ablations are therefore not particularly informative.

---

### Author Response · Authors · 2025-12-02
**Rebuttal Summary**

Dear PCs, SACs, ACs, and reviewers,

Thank you for your consideration. Due to the reviewer data leak, the entire community has been affected. I sincerely appreciate everyone’s efforts in helping to protect and maintain our community. We have addressed the vast majority of the reviewers’ concerns. During the entire rebuttal period, there was no interaction between the reviewers and us. Below is a summary of our rebuttal for your review and assessment.

We are grateful that several of our strengths and contributions were recognized, including the following:

**Strength 1:**
The paper introduces a novel and conceptually simple auxiliary proportional-value classification task to address the learning from label proportions (LLP) setting.

**Strength 2:**
It proposes a divide-and-conquer strategy based on the Fourier transform, which substantially accelerates what would otherwise be a computationally expensive procedure.

**Strength 3:**
Under the assumption that instances within each bag
Bag $B$ are i.i.d., the paper provides strong theoretical support by connecting bag-level proportional-value prediction with instance-level label prediction, and clearly explains why the proposed method can effectively mitigate the over-smoothing problem.

We thank the reviewers for their thoughtful concerns and constructive questions. We have carefully addressed all of the raised issues, and the key points of our responses and resolutions are summarized as follows:

**Problem 1: Extending LLP-PVC to Larger Bag Sizes** raised by reviewers cMjG, QG4B, and B5vC.

In the original manuscript, our bag size settings mainly follow the prevailing setups in multiclass LLP. We additionally evaluated four benchmark datasets with bag sizes extended to 1024, and conducted comprehensive comparisons with the baselines.

**Problem 2: Extending LLP-PVC to More Bag Generation Strategies** raised by reviewers QG4B and B5vC.

The original manuscript mainly focuses on the assumption that the instances $x$ are i.i.d. and then randomly grouped into bags $B$, which is a common setting in LLP. Nevertheless, LLP-PVC can be naturally extended to other LLP scenarios as well. In terms of experiments, we additionally introduced feature-based bag generation for image datasets, as well as a strategy that first determines the label proportions and then constructs the bag $B$. From a theoretical perspective, when the bags $B$ are not i.i.d., as long as the label $y$ is conditionally independent given $x$, our method should be regarded as a heuristic extension. In contrast, for scenarios where the bags $B$ are i.i.d. while the instances $x$ may exhibit dependencies, our method still holds and Theorems 1–3 remain valid.

**Problem 3: Missing Comparisons with Some Baselines or SOTA Methods** raised by reviewers QG4B and cMjG.

We primarily focus on comparing against vanilla algorithms, i.e., methods without additional representation-enhancement or regularization terms. We have additionally included an experiment where we augment LLP-PVC with a regularization term, which achieves SOTA performance. For other baselines that are not included, we provide explanations—for example, some methods are only applicable to binary classification tasks and therefore are not directly comparable to our multi-class setting.

**Problem 4: Fixed Bag Size in Theory and Experiments** raised by reviewers B5vC and QG4B

The difficulty of LLP inherently depends on the bag size: as the bag size increases, the label proportion information becomes less informative at the instance level. We therefore fix the bag size in both our theoretical analysis and experiments to better highlight the effectiveness of our method under different levels of supervision strength in a controlled and comparable setting. If we were to allow varying bag sizes, we would need to impose additional assumptions on the bag-size distribution  which is essentially similar in spirit to assuming a fixed bag size and would not substantially change our conclusions.

**Problem 5: Tightness of Our Bound (bag size in the numerator)** raised by reviewer B5vC

Our error bound depends on the bag size $m$, mainly because we introduce an $m$-way enumeration and an $m$-class surrogate loss. In LLP, it is in fact common for the error bounds to contain $m$ in the numerator, since the difficulty of LLP depends on the bag size: the larger the bag, the more ambiguous the information about individual instances becomes.. Whether our specific dependence on $m$ is optimal or can be further improved remains an open problem.

In response to the reviewer’s request, we conducted an additional experiment: with a fixed  the number of bags, we increased the bag size and did not observe a clear improvement in performance.

The remaining details can be found in the section below, where we provide point-by-point responses to each question. Once again, we sincerely appreciate your valuable review and insightful comments.

---

### Meta-Review · Area_Chair_hYcR · 2026-01-14

**Summary:**

This paper considers the problem of learning classifiers from weakly supervised data (label proportions for a bag of input examples). It addresses the over-smoothing problem using a reformulation of this problem: the label proportions are considered as targets for an auxiliary multi-class classification task. The paper also performed both theoretical and empirical analysis of this approach to demonstrate the main claims.

The reviewers' appreciated the novelty of the formulation and the efficient divide-and-conquer strategy for solving it, but also raised a number of concerns including:
1. Extension to larger bag sizes
2. Extension to other bag generation strategies
3. Missing baseline comparisons
4. Fixed bag size in both theory and experiments
5. Tightness of theoretical bound

The authors' rebuttal addressed #1 (satisfactorily), #2 (mostly), #3 (less satisfactorily), #4 (satisfactorily), and #5 (satisfactorily). It added some newer baselines but could consider expanding them. The main limitation of the method is that it assumes IID bag generation process, also acknowledged by the authors.

I recommend accepting the paper and strongly encourage the authors' to incorporate the rebuttal discussion and address the baselines part in the final paper.

**Reviewer Concerns:**

1. Extension to larger bag sizes
2. Extension to other bag generation strategies
4. Fixed bag size in both theory and experiments
5. Tightness of theoretical bound

are addressed satisfactorily.

3. Missing baseline comparisons

can be further improved to make the evaluation stronger.

**Reviewer Scores:**

Reviewer cMjG: 4 => 6
Reviewer QG4B: 4 => 5
Reviewer B5vC: 4 => 5
Reviewer n3aR: 6 => 7

---

### Decision · Program_Chairs · 2026-01-26

Accept (Poster)